# Beyond Semantics: The Unreasonable Effectiveness of Reasonless Intermediate Tokens

**Karthik Valmeekam**[*][†]                                                   *kvalmeek@asu.edu*
*School of Computing and AI*
*Arizona State University*

**Vardhan Palod**[*]                                                            *vpalod@asu.edu*
*School of Computing and AI*
*Arizona State University*

**Kaya Stechly**[*][‡]                                                          *khstechl@asu.edu*
*School of Computing and AI*
*Arizona State University*

**Atharva Gundawar**                                                        *agundawa@asu.edu*
*School of Computing and AI*
*Arizona State University*

**Subbarao Kambhampati**                                                        *rao@asu.edu*
*School of Computing and AI*
*Arizona State University*

**Reviewed on OpenReview:** *https://openreview.net/forum?id=gDE7YcRC3F*

## Abstract

Recent impressive results from large reasoning models have been interpreted as a triumph of Chain of Thought (CoT), and especially of the process of training on CoTs sampled from base LLMs in order to help find new reasoning patterns. While these traces certainly seem to help the model performance, it is not clear how they actually influence model performance, with some works ascribing semantics to them and others cautioning against relying on them as transparent and faithful proxies of the model's internal computational process. To systematically investigate the role of end-user semantics of derivational traces, we set up a controlled study where we train transformer models from scratch on formally verifiable reasoning traces and the solutions they lead to, constraining both intermediate steps and final outputs to align with those of a formal solver. We notice that, despite significant improvements over the solution-only baseline, models trained on entirely correct traces can still produce invalid reasoning traces even when arriving at correct solutions. More interestingly, our experiments also show that models trained on corrupted traces, whose intermediate reasoning steps bear no relation to the problem they accompany, achieve performance largely comparable to those trained on correct traces. In fact, our corrupted models generalize better on out-of-distribution tasks. We also study the effect of GRPO-based RL post-training on trace validity, noting that while solution accuracy increase, this is not accompanied by any improvements in trace validity. Finally, we examine whether reasoning-trace length reflects *inference-time scaling* and find that trace length is largely agnostic to the underlying computational complexity of the problem being solved. These results challenge the assumption that intermediate tokens or "Chains of Thought" reflect or induce predictable reasoning behaviors and caution against anthropomorphizing such outputs

---

[*]Equal Contribution
[†]Work done while at ASU, currently at Amazon AGI
[‡]Work done while at ASU, currently at Yale University

or over-interpreting them (despite their mostly seemingly forms) as evidence of human-like or algorithmic behaviors in language models.[1]

## 1 Introduction

Recent advances in general planning and problem solving have been spearheaded by so-called "Long Chain-of-Thought" models, most notably DeepSeek's R1 (Guo et al., 2025). These transformer-based large language models, following the now-standard teacher-forced pre-training, instruction fine-tuning, and preference alignment stages, undergo additional training on reasoning tasks. At each step within this additional training, the model, when presented with a question, generates a sequence of intermediate tokens (colloquially or perhaps fancifully called a "Chain of Thought" or "reasoning trace") before producing a specially delimited answer sequence. A formal system then verifies this answer, and the model's parameters are updated to increase the likelihood of generating sequences that result in correct solutions.

While (typically) no optimization pressure is applied to the intermediate tokens (Baker et al., 2025; Zhou et al., 2025), empirically it has been observed that language models perform better on many domains if they output such tokens first (Nye et al., 2021; Wei et al., 2022; Zhang et al., 2022; Hsieh et al., 2023; Gu et al., 2023; Guo et al., 2025; Pfau et al., 2024; Muennighoff et al., 2025; Li et al., 2025). While the fact of the performance increase is well-known, the reasons for it are less clear. Previous work has often framed it in anthropomorphic terms, claiming that these models are "thinking" before outputting their answers (Nye et al., 2021; Gandhi et al., 2025; Guo et al., 2025; Yang et al., 2025a; Zhou et al., 2025; Bubeck et al., 2023; Venhoff et al., 2025).

Famously, DeepSeek's R1 paper claimed that one of the most impressive observed behaviors of their trained models was the so-called "aha" moment: along the way to answering some question, the model output the token "aha", seeming to indicate that it had come upon a sudden realization. Interpreting these tokens as meaningful to the end user requires making an additional, often overlooked assumption about how long CoT models function: that the traces they produce and the traces they were trained on are semantically meaningful to the end user in the same way. While traces certainly seem to help the LLM performance (and may well have mechanistic interpretability properties to better understand the performance improvements (Bogdan et al., 2025)), it is not clear (beyond anecdotal evidence) that they have end-user interpretability or semantics.

For R1 and similar large models, this is nearly impossible to check. The intermediate tokens that massive pre-trained and post-RL'd models produce meander for dozens of pages, are written wholly in ambiguous and polysemantic natural language, and – perhaps much worse – are the result of long, opaque training processes on data that we have no access to and cannot compare against. This has caused significant confusion about their role: some works ascribe semantics to these traces - similar to the "Aha" moment claims in (Guo et al., 2025), authors in Gandhi et al. (2025) claim that four key cognitive behaviors - verification, backtracking, subgoal setting, and backward chaining- are responsible for increasing solution accuracy with traces. In (Marjanović et al., 2025), authors claim that reasoning traces follow a consistent structure of problem formulation followed by repeated cycles exploration and solution verification; others go further, interpreting certain patterns in the traces as evidence of deliberation or even deceptive behavior, raising potential safety concerns (Baker et al., 2025; Korbak et al., 2025; Chua et al., 2025; Greenblatt et al., 2024); while yet another set of works caution against treating these outputs as faithful indicators of the model's internal computation (Chen et al., 2025; Arcuschin et al., 2025).

To cut through this ambiguity, we present a systematic and controlled study designed to investigate the role of intermediate tokens' end-user semantics in transformers. Following previous work that elucidated important functional aspects of large scale models through controlled small scale experiments (Wang et al., 2024; Power et al., 2022; Zhong et al., 2023) and working within a sort of "model organism" paradigm, we focus on fully controlled, open, and replicable models trained from scratch. Our models (0.5B parameter Qwen models) are trained from scratch to solve a simple and well-understood shortest path planning problem

---

[1] 🌐 Project Webpage, 🪧 Code

in grid-based mazes, on training data that includes formally verifiable reasoning traces generated by the A* search algorithm. This setup allows us to systematically evaluate the validity of the traces that models produce at inference time, enabling a clear analysis of the role trace semantics play in model performance. We implement a variety of maze generation algorithms, letting us thoroughly experiment with the training data and measure both in and out of distribution performance.

Using this framework, we structure our investigation around a series of core questions. We begin by examining the most fundamental assumption: the relationship between a correct solution and the validity of the reasoning trace that precedes it. Specifically, we ask whether the correctness of a generated solution correlates with the validity of its intermediate tokens. If these tokens are semantically meaningful with respect to the problem, correct answers should always come from valid traces, and valid traces should always lead to correct answers. However, our findings reveal that this connection is weaker than that—models often produce correct solutions despite invalid reasoning traces. We next examine whether the semantic quality of traces used during training influences model performance. To test this, we train models on deliberately corrupted datasets which swap reasoning traces between problems, removing any connection to the task at hand. Recognizing that many state-of-the-art models are post-trained with reinforcement learning from verifiable rewards, we further investigate whether such techniques (like Group Relative Policy Optimization (GRPO) (Shao et al., 2024)) improve trace validity in addition to solution accuracy. Finally, we address the popular notion that longer traces represent a form of "inference-time scaling" or problem-adaptive computation by analyzing whether the length of a generated trace correlates with the problem's underlying computational complexity.

Our findings consistently challenge the prevailing narrative that intermediate tokens constitute a semantically meaningful reasoning process. First, we observe a pronounced lack of correlation between solution correctness and trace validity—models frequently produce invalid reasoning traces even when they arrive at correct solutions. Second, and more strikingly, models trained on corrupted or semantically irrelevant traces achieve performance comparable to, and often exceeding, that of models trained on correct traces, especially on out-of-distribution tasks. Third, although post-training with reinforcement learning improves solution accuracy across both in- and out-of-distribution settings, it does not consistently enhance trace validity. In fact, we find cases where reinforcement learning decreases trace validity while simultaneously improving solution accuracy for models trained on correct traces. Moreover, models trained on corrupted traces continue to outperform their correct-trace counterparts across domains while consistently generating invalid reasoning traces. Finally, we find that the length of the generated traces is largely agnostic to the difficulty of the underlying problem, undermining the notion that it reflects problem-adaptive computation.

Together, these results suggest that the effectiveness of intermediate tokens does not arise from their seemingly interpretable semantic content. By systematically disentangling trace semantics from the underlying problem, our study demonstrates that if performance is the objective, assuming human-like or algorithmically interpretable trace semantics are ideal or even achievable is not only unnecessary but potentially misleading.

## 2 Related Work

**Post-Training for Reasoning -** Recent progress in improving the reasoning capabilities of Large Language Models (LLMs) has been driven by methods that train models not only on correct answers, but also on "reasoning traces" that lead to those answers (Zelikman et al., 2022; Li et al., 2025; Muennighoff et al., 2025; Lightman et al., 2023; Lambert et al., 2024; Yuan et al., 2024; Guo et al., 2025; Sun, 2023; Guo et al., 2025; Arora & Zanette; Yu et al., 2025). Whether this is achieved via supervised fine-tuning on trace-augmented datasets or via reinforcement learning techniques like Group Relative Policy Optimization (GRPO), the end result is the same: models "think" for varying amounts of time by outputting additional tokens before their final answers, and performance measures improve. While these papers demonstrated ways to improve final answer accuracy, they neither evaluate the trace accuracy nor do they explicitly attempt to train on incorrect or irrelevant traces. Thus, they leave open the question of whether that accuracy increase actually stems from the additional semantic information in the trace.

These methods are often paired with claims about how and why they work, which lean on anthropomorphic framings of model "thinking", which seem to assume that final answer correctness somehow requires intermediate sequence correctness. These claims are even presented alongside evidence that seems to directly

contradict them. The training procedure underlying DeepSeek's R1-Zero is instructive (Guo et al., 2025): it pays attention only to the correctness of the solution, ignoring the content of the traces. Incorrect traces that happen to lead to correct answers are treated exactly the same as correct ones, and their ablations hint that rewarding intermediate correctness might even be counterproductive. In this work, we push this idea further and systematically evaluate how the final performance is affected when explicitly incorrect traces are rewarded.

**Evaluating Traces -** Previous work on evaluating the relationship between trace correctness and final answer correctness has primarily focused on large, pre-trained models, and has gone under the name of Chain-of-Thought faithfulness. These evaluations have claimed that intermediate steps, despite appearing coherent, are not reliably correct (Turpin et al., 2023; Lanham et al., 2023; Chen et al., 2025; Chua & Evans, 2025; Arcuschin et al., 2025; Baker et al., 2025). However, the nature of natural language makes these evaluations questionable. Within the domain of math problems, there is no established ground truth semantics for what a correct reasoning process looks like in natural language, and so previous work either relies on messy manual evaluations which typically require the researcher to read in some amount of intent into the model's completions, or automated evaluations that use additional (unverified) language models to noisily pick out potential reasoning errors.

**Neural Algorithmic Reasoning and Training Transformers on Traces -** There have been works in Neural Algorithmic Reasoning (NAR) (Veličković & Blundell, 2021) which focused on inducing faithful representation of classical algorithms in Neural Networks (Veličković et al., 2022; Li et al., 2024). These studies have often analyzed whether the intermediate computations in the Networks correspond to a faithful representation of the ground truth algorithm (Veličković et al., 2019). However, existing works in NAR have focused on encoding graph algorithms in variants of Graph Neural Networks (Veličković et al., 2017; 2020; Brody et al., 2021; Saldyt & Kambhampati, 2025) rather than autoregressive transformers.

For transformer models, prior efforts have trained models from scratch to mimic search algorithms like A*, Breadth-First Search (BFS), and Monte Carlo Tree Search for tasks in pathfinding, arithmetic, and general problem-solving (Lehnert et al., 2024; Gandhi et al., 2024; Yang et al., 2022), as well as the DPLL procedure for Boolean SAT problems (Pan et al., 2024). However, while these studies train on traces of formal procedures, they do not explicitly analyze the correctness of the traces generated by the final model. While some works happen to use semantically invalid traces—such as truncated A* derivations in Dualformer (Su et al., 2024)—they do not examine how they affect model performance. By contrast, our work is the first to rigorously evaluate trace correctness in autoregressive transformer models, allowing us to directly investigate the relationship between generated solutions and the tokens that led to them.

## 3 Background

In many cases, especially with pre-trained models, it is nearly impossible to formally verify reasoning traces, due to the ambiguity of natural language and the lack of a clear ground truth.[2] However, for small, well-scoped formal domains like the gridworld path planning domain used in (Lehnert et al., 2024; Su et al., 2024) and this paper, and by carefully training models from scratch on those domains, we have the ability to check whether generated traces follow the exact semantics enforced in the training data and causally predict the final solutions that the model outputs.

### 3.1 The Maze Pathfinding Domain

We consider a standard grid-based path-finding domain. The task is to find a legal path between a given start cell and goal cell in a $30 \times 30$ grid. Every cell of this grid is either free (traversable) or a wall (impassable). The agent begins at the start state, and at every state may take one of four actions: go up, down, left, or right. The transformer is given a full description of this problem (in token format – we follow the formulation used by (Lehnert et al., 2024) and (Su et al., 2024)) and must output as its final answer a plan, which consists

---

[2]This in turn induces a Rorschach-test-like behavior in the end users who overload semantics on specific phrases like "aha" or "let me think".

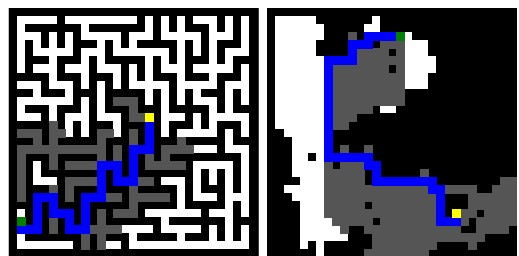

Figure 1: Examples of mazes. The left is generated by Wilson's algorithm and is used for model training. The right is generated by the Drunkard's Walk algorithm and used to evaluate models on an out of distribution task. The goal is represented by a green square and the start state by a yellow square. Black squares represent impassable walls. Blue squares represent steps along the optimal path (as found by A*). Gray squares are squares that were explored by A* but are not along the optimal path. White squares are unexplored traversable squares.

of a sequence of actions. A plan is considered correct if every action the agent from a free cell to an adjacent free cell and if it results in the agent's final position being the goal cell.

In order to understand out of distribution performance, we generate navigation problems using diverse generation algorithms, resulting in varied structural patterns and exploration dynamics. This enables systematic out-of-distribution (OOD) evaluation by testing models on maze types unseen during training. These generation algorithms can be sorted into two major categories: 1) algorithms that do not permit cycles and sample spanning tree mazes embedded in the $30 \times 30$ grid and 2) algorithms that permit loops and create noisy, less-structured dungeon or cave-like instances. For all algorithms except Searchformer's (see below), which has its own start and goal generation loop, we sample a legal (start, goal) pair after maze generation.

**Acyclic Maze Generation**

1. **Wilson's algorithm:** Wilson's algorithm generates uniform random mazes by performing loop-erased random walks from unvisited cells until they connect to the current maze (Wilson, 1996). Each walk removes any loops it creates, ensuring a valid tree structure. This process continues until all cells are included, producing a uniform sample from the space of all possible spanning trees of a $15 \times 15$ square lattice graph. As there are two choices for embedding such a maze into a gridworld (either a $(0,0)$ or $(0,1)$ offset from the top left), we select between them uniformly. We use this same choice for all algorithms in this category.

2. **Kruskal's algorithm:** Kruskal's algorithm, originally proposed for finding a minimum spanning forest of an undirected edge-weighted graph (Kruskal, 1956), generates mazes by treating each cell as a node and randomly removing walls between unconnected regions, using a union–find structure to avoid cycles. This results in a fully connected maze without loops, though the maze distribution is not perfectly uniform. The method produces mazes biased towards short local connections and dead ends.

3. **Randomized Depth-First Search algorithm (DFS):** The randomized depth-first search (DFS) or recursive backtracker algorithm generates mazes by carving a path forward until reaching a dead-end (Tarjan, 1972). When it hits a dead-end (no unvisited neighbors), it backtracks until it finds a new direction to explore, repeating until all cells are visited and connected into a complete maze. Depth-first search is biased towards generating mazes with low branching factors and many long corridors.

**Cave Generation**

4. **Drunkard's Walk:** We implement a version of the "Drunkard's Walk" algorithm, as described by (Heard, 2016), and originally used for procedurally generating dungeons for top-down two-dimensional

video games. Starting from a grid of solid walls, a random walk is performed, carving out the current cell on every step. The walk continues until a predefined number or percentage of floor tiles has been dug out. This method allows cycles, producing cave-like structures with open chambers and looping corridors. The output space includes grid states unreachable by perfect acyclic maze generators.

5. **Searchformer style generation (SF-style)** We also implement the random generation algorithm used in the Searchformer paper (Lehnert et al., 2024). Tasks are generated by exhaustive rejection sampling: first, a random wall density between 30% and 50% is selected. That proportion of cells is then designated as walls. A start and goal location are sampled, and A* search is executed to compute an optimal plan. Unsolvable, trivial, or duplicate instances are rejected and resampled. These mazes may contain loops and are therefore out of distribution relative to the acyclic training sets.

In our experiments, we train two separate models: one on datasets generated using an acyclic maze-generation algorithm (Wilson's) and another on datasets generated using a cave-style algorithm (SF-style).

### 3.2 The A* Search Algorithm

A* is a classic best-first graph–search procedure that combines the uniform-cost guarantee of Dijkstra's algorithm (Dijkstra, 1959) with domain-specific heuristics to focus exploration on promising states, originally introduced to compute minimum-cost paths in state-space graphs (Hart et al., 1968).

The algorithm maintains an *open list* (a priority queue) keyed by $f(n) = g(n) + h(n)$, where $g(n)$ is the exact cost from the start and $h(n)$ is a heuristic estimate to the goal, and also maintains a *closed list* of already visited nodes. It repeatedly pops the open list node with the smallest $f$; if this is the goal, it reconstructs the path that lead to this node and this is returned as the final plan. Otherwise, it generates child nodes (in our case, traversable neighbor cells) and calculates their $g$ and $f$ values. For each node, it either inserts it into the open list or – if the node is already in the list – updates its $g$ value if the new value is lower. The popped node is added to the closed list to prevent re-expansion.

The effectiveness of A* is dependent on the heuristic it is implemented with. For solvable graph search problems like the ones featured in this paper, any consistent $(h(n) \leq c(n, n') + h(n')$ for all neighboring $n')$ heuristic will guarantee that the plan returned is not only satisficing but optimal (Pearl, 1984).

For the maze path planning problems we examine in this paper, we use the very standard Manhattan heuristic $h(n) = |x_n - x_g| + |y_n - y_g|$ which computes the sum of horizontal and vertical displacements between a cell and the goal. On a 2-D grid with only orthogonal, unit-cost movement, this heuristic is consistent, ensuring A* returns an optimal path.

Finally, following Searchformer and Stream of Search, we modify the A* implementation to output a linearized execution trace (Gandhi et al., 2024; Lehnert et al., 2024). That is, whenever the algorithm creates a child node and adds it to the open list, it prints `create x y cG cH` and when it closes a node and adds it to the closed list, it prints `close x y cG cH`. Here, 'x' and 'y' are cell coordinates, 'G' in cG represents the exact cost from the start state to the node (i.e., the $g(n)$ value) and 'H' in cH represents the heuristic estimate from that node to the goal state (i.e., the $h(n)$ value). Similar to Searchformer notation (Lehnert et al., 2024), we use the prefix "c" to differentiate between the node co-ordinates and its cost estimations in our human-readable gloss of the token representation. In the next section, we construct an A* validator that reverses this process – it takes in a linearized trace and attempts to simulate the corresponding open and closed list operations to check if they are valid with respect to the semantics of this implementation.

## 4 Validating Traces and Solutions

While previous work evaluated the final accuracy of trace-trained models, it did not evaluate the traces themselves. For large, production ready RL-post-trained models like DeepSeek's R1, this is practically impossible. For even a simple query, the model produces dozens of pages of convoluted and meandering

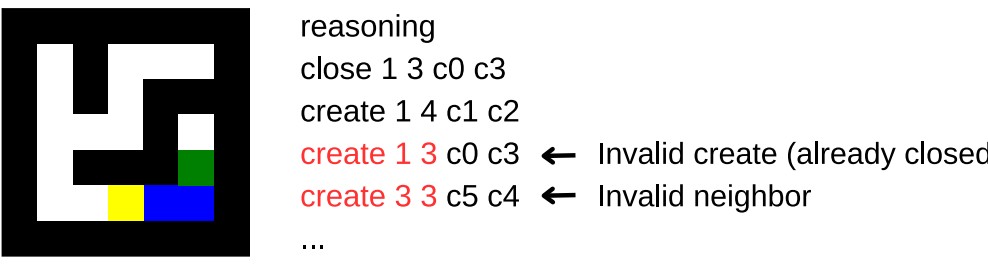

Figure 2: Trace validation procedure. The figure has an example maze problem and a model generated trace. The maze problem, with left bottom corner as $(0,0)$, has start state as (1,3) and goal state as (2,5). Our A* validator runs through the model's output stream sequentially. Assuming no parsing errors, it will flag a trace as invalid if at some point it contains an invalid action.

output before arriving at an answer, and this output is all in natural language, making it very easy to read multiple equally valid interpretations into it.

To truly tell whether the traces that were trained on helped in the expected way, we need a formal way of validating their correctness. By training models on traces produced by a well-known algorithm with well-known semantics, it is possible to check whether the model's emulations of the algorithm's execution trace are correct.

We construct a formal verifier for A* traces. The format for these traces follows (Lehnert et al., 2024), and is described in more detail in Section 3. Essentially, our verifier consumes the generated trace and simulates the operations proposed in that trace on open and closed lists. It runs through the generated trace sequentially, parsing each `action x y cA cB` sequence as an operation and using it to update its open and closed list. It marks a trace as valid if it can correctly execute this procedure until it closes the goal node. Errors in execution can be one of the following:

- **Parsing Error**: a substring is malformed and does not parse into either a create or a close action with the correct arguments.

- **Wall exploration**: the current create/close action is attempting to reference a wall cell.

- **Invalid Neighbor**: the current create action is attempting to create an illegal child, either referencing a wall cell or a cell that is not adjacent to the last closed node.

- **Already Closed**: the current create action is attempting to close an already closed node.

- **Not in Open List**: the current close action is referencing a node that is not in the open list.

- **Trace - Plan mismatch**: the generated plan is different from the plan that is extracted by executing the operations provided in the reasoning trace.

- **Goal Not Reached**: after the entire sequence was processed, the goal node was not in the closed list, and so the reconstruction step cannot proceed.

## 5 Evaluating Solution Accuracy vs. Trace Validity

With this verifier in hand, we can now evaluate the correlation between the trace validity and solution correctness for models trained on a dataset containing correct traces paired with correct plans. To conduct this analysis, we train transformer models from scratch using a controlled architecture and dataset configuration. Specifically, we modify the architecture of the Qwen2.5 0.5B (Qwen Team, 2024) to support a vocabulary of exactly 944 different tokens, reducing the parameter count from 500 million to about 380 million. We randomly initialize the models, and then train it with a batch size of 8 on two NVIDIA H100s. The models have a context length of 32,000 tokens to support the long lengths of intermediate token generation. Our other

experiments later in the paper also use this architecture, but train on different datasets, from solution-only through to irrelevant traces. All code and data will be publicly released. Additional hyperparameter details are provided in Appendix A.1.

Table 1: Performance of Wilson and Searchformer-style Models across maze distributions for models trained on 500k datapoints. Each cell shows *Plan Validity (%) / Trace Validity within valid plans (%)*. "Wilson model" is trained on problems generated using Wilson's algorithm; "SF-style model" is trained on problems generated using Searchformer-style generation.

| Model Type | Wilson | Kruskal | DFS | Drunkard | SF-style |
|---|---|---|---|---|---|
| Wilson Model | 79.9 / 97.7 | 83.2 / 97.2 | 54.0 / 92.0 | 0.0 / - | 0.0 / - |
| SF-style Model | 40.8 / 89.0 | 44.6 / 93.0 | 19.9 / 85.4 | 62.1 / 85.2 | 56.2 / 81.1 |

We train models on two datasets, each containing 500k problems generated using Wilson's algorithm and SF-style generation. Each datapoint consists of a start and goal state, a maze definition, an A$^*$ trace representing the search process between the given start and goal states, and the corresponding correct plan. We evaluate these models on testsets generated by Wilson, Kruskal, DFS, Drunkard and SF-Style maze generation algorithms. Each test dataset consists of 1000 instances. Model performance is measured in terms of solution accuracy, and we also evaluate trace validity for responses that yield correct solutions. Note that we do not check for optimality in either the traces or the plans.

We focus on the metric *Trace Validity within valid plans (%)* as we wanted to provide a formally verifiable trace validity claim. Prior work that has looked at trace evaluations has only focused on partial and qualitative metrics in natural language settings (Marjanović et al., 2025; Gandhi et al., 2025; Samineni et al., 2025). In contrast, here we wanted to provide guarantees regarding the global validity of intermediate tokens. This stringent notion of trace correctness is directly aligned with the core question of whether trace correctness is tightly coupled to correct solutions.

As shown in Table 1, the results highlight that the trace validity in a response is not truly a reliable predictor of plan accuracy: if it were, the trace validity within valid plans would consistently reach 100%.[3] This discrepancy is especially pronounced in the SF-style model.

Ideally, if a network has learned an algorithm, this should endow it with the generalization properties of that algorithm. Following this intuition, we further investigate the relationship between solution correctness and trace validity by conducting a controlled-length generalization experiment using models trained on problems generated with Wilson's algorithm. We define problem difficulty as the number of operations required by the A* algorithm to solve a given problem, where longer traces indicate greater difficulty.

The training set consists of 50k problems with solution traces ranging from 1000 to 3500 tokens. To balance the distribution, the dataset was divided into 500-token intervals, with 10k problems in each bin (1000–1500, 1500–2000, 2000–2500, 2500–3000, and 3000–3500).

For evaluation, we design a setup that allows us to probe both easier and harder cases outside the training distribution, as well as held-out problems from within the training range. We construct four disjoint test sets designed to probe model performance across varying problem difficulties. The Easier (0–1000 tokens) and In-distribution held-out (1000–3500 tokens) sets each contain 1000 problems. To obtain higher resolution in the difficult out-of-distribution regime, we split the harder range (3500–4500 tokens) into two subsets of 500 problems each: Harder-1 (3500–4000 tokens) and Harder-2 (4000–4500 tokens).

The purpose of actually learning these traces is to gain the benefits of algorithmic generalization. Yet, as shown in Table 2, the model internalizes the style of the algorithm without acquiring its underlying mechanisms. This gap is also reflected in the deteriorating correlation between trace validity and solution correctness as difficulty increases. In particular, trace validity among correct solutions further decreases on

---

[3]In the appendix, we provide a complete breakdown of model responses as confusion matrices.

Table 2: Length generalization results for the Wilson Model. Test bins are grouped into easier, in-distribution, and harder categories based on trace length ranges. "Plan Acc." = Plan Accuracy, "Trace Val. in Val. Plans" = Trace Validity within valid plans.

| Category | Plan Acc. (%) | Trace Val. in Val. Plans (%) |
|---|---|---|
| Easier (0–1000) | 72.8 | 93.2 |
| In-distribution held-out (1000–3500) | 67.8 | 87.9 |
| Harder-1 (3500–4000) | 35.0 | 61.7 |
| Harder-2 (4000–4500) | 16.2 | 51.8 |

longer traces, suggesting that the correlation is not a fundamental property but rather an artifact of the training distribution.[4]

## 6 Training with Traces: Does Meaning Matter?

If plan and trace validity are only loosely connected for models trained on the A* trace dataset, then perhaps the validity of the trace isn't as important to the performance increase as previously believed. To test this empirically, we construct a second training dataset called *Swapped*, which we build by randomly permuting reasoning traces between problems.

We construct Swapped datasets from two of the original datasets: the Wilson's algorithm set and the SF-style one. These datasets consists of the exact same problems as the original 500k datasets, but problem 1's trace will be given to, say, problem 4; problem 4's will be given to problem 7; and so forth. In other words, while these traces continue to have the right form and some generic domain information, they no longer have any connection to the specific problems they are associated with. Training examples consist of a start and goal state, a maze definition, an A* trace for searching for the shortest path across a totally unrelated maze from a different start and goal state, and the correct solution plan for the original maze.

For these experiments, we continue to use the same model architecture described in the previous section, varying only the datasets we train on to see how they affect performance – even when the reasoning traces are entirely disconnected from the underlying problem.

Table 3 provides accuracy/validity results over a spectrum of training runs. The most basic training run is the standard solution-only baseline, where the model is trained on just solutions without any derivational traces. The next baseline, following previous work (Lehnert et al., 2024; Su et al., 2024; Gandhi et al., 2024), is training the models with A* generated traces, teacher-forcing during training to make it output intermediate tokens before the final solution. These correspond to the "Normal" models discussed in the previous section. Finally, we train models on the Swapped datasets where the intermediate trace is completely unrelated to the problem being solved. Each model is trained on 500K samples and evaluated on tests sets containing 1000 problems each, generated using multiple maze generation algorithms (as described in Section 3).

As seen in Table 3, the swapped models, which we might expect would perform worse than the normal ones, counterintuitively perform just as well, if not better than the model trained on correct traces and substantially better than the solution-only baseline! We see that they have 0% trace validity in all cases–as expected given that they have been trained with well-strutured but problem-irrelevant traces.

The performance difference is particularly striking on out-of-distribution datasets. For Wilson models, while most of the performance gaps are within a few percentage points-and in-distribution testing results in near identical performance-the swapped model achieves 11.7% plan accuracy on the Drunkard dataset, whereas the normal model fails to solve any problem. Similarly, for models trained on SF-style data, the swapped model performs much better than the normal model across all distributions. In particular, the swapped model achieves a plan accuracy of 95.4% on Drunkards as compared to the 62.1% accuracy of the normal

---

[4]In Appendix A.5, we also show that the original Searchformer model, as trained by (Lehnert et al., 2024), also exhibits this disconnect between the trace validity and solution accuracy, even though those authors didn't seem to have evaluated this.

Table 3: Performance of Solution-only baseline, Wilson-trained model, and Searchformer-style (SF-style) trained model across maze distributions (final checkpoint shown). Each cell shows *Plan Accuracy (%) / Trace Validity within valid plans (%)*. For the Solution-only baseline, only Plan Validity is reported.

| Model Type | Wilson | Kruskal | DFS | Drunkard | SF-style |
|---|---|---|---|---|---|
| Solution-only (Wilson) | 10.0 / − | 6.9 / − | 0.0 / − | 0.1 / − | 0.0 / − |
| Normal (Wilson) | 79.9 / 97.7 | 83.2 / 97.2 | 54.0 / 92.0 | 0.0 / 0.0 | 0.0 / 0.0 |
| Swapped (Wilson) | 83.3 / 0.0 | 85.0 / 0.0 | 52.6 / 0.0 | 11.7 / 0.0 | 0.2 / 0.0 |
| Normal (SF-style) | 40.8 / 89.0 | 44.6 / 93.0 | 19.9 / 85.4 | 62.1 / 85.2 | 56.2 / 81.1 |
| Swapped (SF-style) | 45.8 / 0.0 | 51.0 / 0.0 | 29.0 / 0.0 | 95.4 / 0.0 | 89.1 / 0.0 |

model. We provide results of additional training runs and a detailed error-wise breakdown of model generated responses in the Appendix sections A.2, A.3 and A.4.1. We also did experiments on pre-trained models, despite being aware that there are major confounding factors such as unknown pre-training data and training methodologies. We finetuned Qwen3 8B model across the 3 different training paradigms (details provided in the Appendix section A.6) We find that the results are largely consistent with the results reported in Table 3. In a companion work which focused on pretrained models fine-tuned on traces in the Question Answering (QA) domain, similar results have been observed (Bhambri et al., 2026).

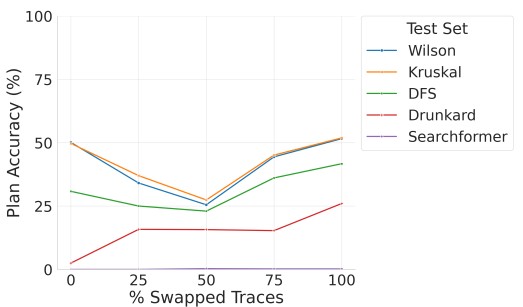

(a) % of swapped traces vs plan accuracy.

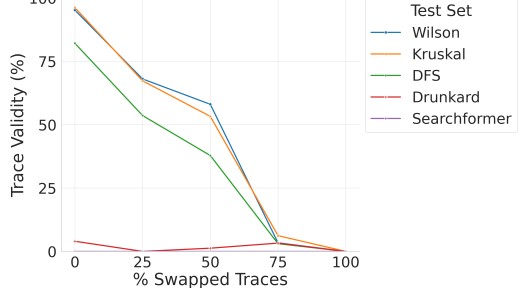

(b) % of swapped traces vs trace validity among valid plans

Figure 3: Effect of increasing percentage of swapped traces in the training dataset (Wilson mazes with 50k samples).

We conducted an additional study varying the proportion of swapped traces during training on a 50k sample Wilson dataset. Instead of fully replacing the trace distribution, we generated training sets where only a fraction of examples received mismatched traces, allowing us to interpolate between the normal and fully swapped settings. As shown in Figure 3a, we observe a U-shaped performance trend: introducing a small amount of trace swapping initially reduces plan accuracy, but beyond a certain point, plan accuracy begins to rise again—eventually matching or even exceeding the normal model—while trace validity steadily decreases as we move toward the fully swapped setting (in Figure 3b).[5] This pattern suggests that what matters for transformers to improve accuracy with intermediate tokens is not their semantic correctness, but rather their *consistency*.

If intermediate tokens improve accuracy because they teach the model a given reasoning procedure, then we should expect their influence on performance to worsen as they become disconnected to the problem. However, we find that this is not always the case – in fact, intermediate token sequences that have almost nothing to do with the problem at hand can provide a significantly higher performance boost (and which, counterintuitively might even generalize better) than well-grounded semantically meaningful execution traces, thus throwing doubt on the seemingly wide-spread intuition that the effectiveness of traces stems from allowing the

---

[5]We also show in Appendix A.7 that post-training can further boost these models' performance without inducing trace validity!

transformer to perform relevant, interpretable, algorithmic procedures. Our results suggest that while traces are helping model accuracy, this boost is not connected with the expected end-user semantics.

### 6.1 Does post-training increase the semantic correctness of the intermediate tokens?

Table 4: Performance of Normal and Swapped (Wilson) models across maze distributions for each checkpoint during GRPO training. Each cell shows *Plan Accuracy / Trace Validity within Valid Plans (%)*.

| Checkpoint | Model Type | Wilson | Kruskal | DFS | Drunkard | SF-style |
|---|---|---|---|---|---|---|
| 0 | Normal | 79.9 / 97.1 | 83.1 / 95.6 | 54.0 / 87.4 | 0.2 / 0.0 | 0.0 / - |
|   | Swapped | 83.3 / 0.0 | 85.0 / 0.0 | 52.6 / 0.0 | 11.7 / 0.0 | 0.2 / 0.0 |
| 70 | Normal | 88.9 / 97.9 | 89.1 / 98.1 | 62.3 / 92.1 | 0.0 / - | 0.0 / - |
|   | Swapped | 93.4 / 0.0 | 93.5 / 0.0 | 67.1 / 0.0 | 14.4 / 0.0 | 0.4 / 0.0 |
| 140 | Normal | 89.6 / 98.2 | 91.7 / 98.2 | 62.4 / 92.9 | 0.0 / - | 0.0 / - |
|   | Swapped | 99.5 / 0.0 | 99.3 / 0.0 | 81.9 / 0.0 | 14.4 / 0.0 | 0.4 / 0.0 |

Table 5: Performance of Normal and Swapped (Searchformer) models across maze distributions for each checkpoint during GRPO training. Each cell shows *Plan Accuracy / Trace Validity (%)*.

| Checkpoint | Model Type | Wilson | Kruskal | DFS | Drunkard | SF-style |
|---|---|---|---|---|---|---|
| 0 | Normal | 40.8 / 89.0 | 44.6 / 93.0 | 19.9 / 85.4 | 62.1 / 85.2 | 56.2 / 81.1 |
|   | Swapped | 45.8 / 0.0 | 51.0 / 0.0 | 29.0 / 0.0 | 95.4 / 0.0 | 89.1 / 0.0 |
| 70 | Normal | 54.2 / 72.0 | 58.8 / 67.7 | 20.7 / 68.6 | 88.0 / 66.9 | 82.4 / 65.3 |
|   | Swapped | 62.3 / 0.0 | 67.5 / 0.0 | 39.4 / 0.0 | 99.6 / 0.0 | 96.9 / 0.0 |
| 140 | Normal | 63.7 / 52.1 | 68.8 / 49.9 | 27.9 / 50.5 | 91.2 / 45.7 | 89.1 / 43.4 |
|   | Swapped | 69.0 / 0.0 | 75.5 / 0.0 | 41.6 / 0.0 | 99.2 / 0.0 | 98.5 / 0.0 |

Considering that claims anthropomorphizing the relationship between intermediate tokens and the solution (such as the "aha" moment) (Nye et al., 2021; Gandhi et al., 2025; Guo et al., 2025; Yang et al., 2025a; Zhou et al., 2025) originate from models that are post-trained with reinforcement learning methods (like Group Relative Policy Optimization (GRPO)), we investigate whether such post-training improves the semantic correctness of the traces.

We use the models described in the previous section (trained on 500k samples for 255k steps) as base models and further post-train them on separate 10k-sample datasets. Specifically, the Wilson model is post-trained on a distinct Wilson dataset, and the SF-style model is post-trained on a distinct SF-style dataset. Additional hyperparameter details are provided in Appendix A.1.

As shown in Tables 4 and 5, GRPO improves solution accuracy across both in- and out-of-distribution settings but does not consistently enhance trace validity. Notably, in several cases, post-training even reduces trace validity while simultaneously improving solution accuracy for models trained on correct traces. For the SF-style normal models (shown in Table 5), trace validity steadily decreases—from values in the mid-80% range to the mid-50% range—while their plan accuracy increases substantially across nearly all distributions. Furthermore, the swapped models continue to outperform their correct-trace counterparts across all distributions, despite maintaining a consistent trace validity of 0%. We provide a detailed error-wise breakdown of model generated responses across various checkpoints and test sets in Appendix A.4.1. Taken together, these results provide further evidence that user-expected semantics of intermediate tokens are not the main driver of performance improvement.

## 7 Is the length of Intermediate Tokens a true reflection of Problem Complexity?

Along with the semantics of intermediate tokens, prior work has often anthropomorphized trace length itself—interpreting longer reasoning traces as evidence that models are "thinking harder" on more complex problems (Guo et al., 2025; Chen et al., 2024; Yang et al., 2025b; Muennighoff et al., 2025). This interpretation assumes that models dynamically allocate more computation when faced with difficult inputs, mirroring problem-adaptive reasoning. Consequently, producing longer reasoning traces has been described as *inference-time scaling.* However, such claims are largely unverified: length may instead reflect arbitrary sampling behavior or artifacts of the training procedure, rather than genuine computational adaptivity. To test this hypothesis, we investigate whether the length of generated reasoning traces in our controlled maze navigation setting correlates with the problem difficulty of the underlying problems (Palod et al., 2025).

We quantify problem difficulty in our pathfinding domain by measuring the number of operations performed by the $A^*$ search algorithm to generate the optimal plan—that is, the length of the $A^*$ generated trace. This metric provides a clear and objective measure of relative problem difficulty. We then analyze the lengths of reasoning traces produced by the Wilson and SF-style models on the same test sets described in previous sections. Figure 4 plots the model-generated trace lengths against the corresponding ground-truth $A^*$ trace lengths for both models.

The highly dispersed scatter plots in Figure 4 show that there is no real correlation between model-generated trace and ground-truth $A^*$ trace lengths. For the Wilson models, the Pearson correlation is negligible $r = 0.0081$ ($p = 0.566$), while the Spearman rank correlation shows a tenuous monotonic relationship $\rho = 0.413$ ($p < 0.001$). Similarly, the SF-style model generated traces exhibit weak alignment with $A^*$ search traces , with a Pearson correlation of $r = 0.197$ ($p < 0.001$) and a Spearman correlation of $\rho = 0.509$ ($p \approx 0$). If such a correlation existed, the generated trace lengths would cluster closely around the ground-truth values. In many cases, the model continues generating intermediate tokens up to the maximum context limit of 32k without ever producing a valid solution.

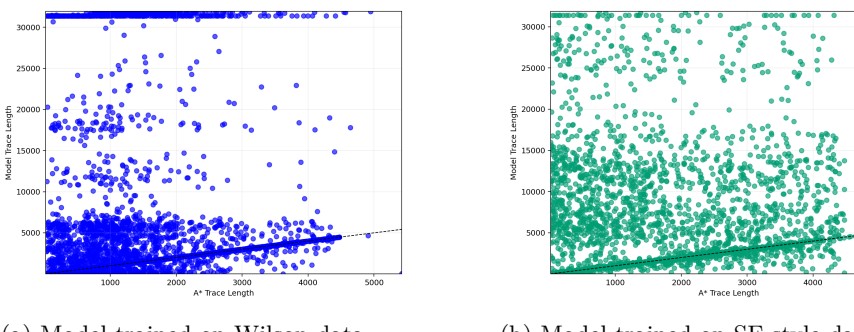

(a) Model trained on Wilson data    (b) Model trained on SF-style data

Figure 4: Comparison of generated intermediate token length (Y-axis) with ground-truth A* trace length (X-axis). The dashed line represents y = x.

We further analyze whether the model's behavior differs between in-distribution and out-of-distribution (OOD) problems, focusing on the model trained on Wilson-generated data. Because this model was trained exclusively on Wilson mazes, it approximates $A^*$-like trace lengths when evaluated on similar mazes, producing a visible alignment between the generated and ground-truth lengths (shown in Figure 5a). This alignment is reflected in a moderate Pearson correlation $r = 0.40$ ($p < .001$) and an high Spearman rank correlation $\rho = 0.98$ ($p \approx 0$). However, when evaluated on SearchFormer-style mazes (Lehnert et al., 2024), this correlation disappears entirely (shown in Figure 5b). In these OOD settings, the generated trace lengths fluctuate independently of problem complexity, reflected by the very low Pearson correlation $r = 0.02$ ($p = .518$) and the Spearmann correlation $\rho = 0.02$ ($p = .550$). These results suggest that the apparent problem-adaptive computation observed for Wilson mazes might be an artifact of the training distribution rather than genuine adaptivity.

In addition to these experiments, in (Palod et al., 2025), we also report experiments where the model trained on Wilson mazes was tested on 100 trivially simple "no-maze" instances (where essentially, the start and goal

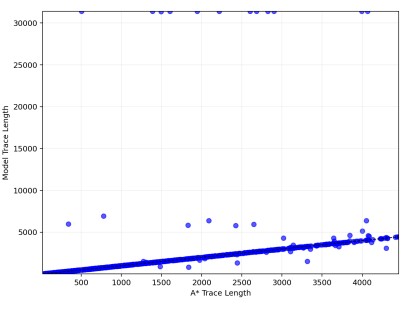

(a) Wilson trace length scatter

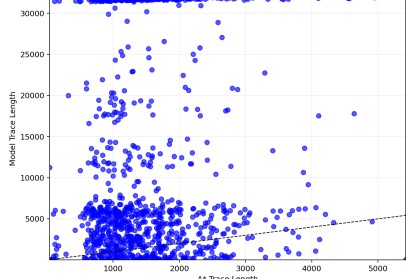

(b) Searchformer trace length scatter

Figure 5: Comparison of trace length scatter plots for problems generated using Wilson and Searchformer Algorithms.

states are in free space). We see that the model gives correct solutions only to 5 out of 100 instances, and in many cases generates excessively long intermediate traces (exceeding the 32K context limit before failing). This further bolsters the conclusion that the intermediate token length is more correlated to the instance being out of distribution, rather than its computational complexity.

## 8 Conclusion and Implications

In this paper, we conducted a systematic controlled study focused on the end-user semantics of intermediate traces in transformers. As we don't have access to any frontier LLM's training data or even exact training procedure, and since the traces these models output are in multiply-interpretable natural language without a concrete ground truth, we designed a controlled study building on previous smaller model reasoning work – mainly Searchformer and Stream of Search (Gandhi et al., 2024; Lehnert et al., 2024). Our findings demonstrate that trace validity in a response is not a reliable predictor of solution correctness. We then found that trace semantics is not necessary for achieving high task performance by training models on traces that are not connected to the problem at hand. Our post-training results show that RL with GRPO improves the performance of the base model, irrespective of the trace validity of the intermediate tokens. Finally, we show that the trace length generated by the models are agnostic to the computational complexity of the problem, which challenges the prevailing narrative that derivational traces reflect problem-adaptive computation.

### 8.1 Implications

All together, our counter-intuitive results demonstrate ways in which common interpretations of the intermediate tokens produced by Large Reasoning Models may be anthropomorphizations or simplifications. Based on these results, we argue that while the traces certainly seem to help the LLM performance (and may well have mechanistic interpretability (Bogdan et al., 2025)), intermediate tokens do not have end-user interpretability/semantics and, if the goal is to increase model performance, enforcing trace semantics is unnecessary and potentially very misleading.

We note that our position is not that intermediate tokens *can't ever have* interpretable meaning but that *they don't necessarily have to have* any interpretable meaning. Any human interpretable meaning in the intermediate tokens might be a fortuitous coincidence of such rationales being present in the training data, rather than the model actually doing internal computations corresponding to those statements. In other words, the correlation here may be spurious rather than causal. This conclusion is also supported to some extent by our companion work with pre-trained models finetuned on Q&A scenarios (Bhambri et al., 2025; 2026).

Our results also raise questions about both the fears of deception in reasoning models based on their chains of thought (Greenblatt et al., 2024), as well as the earnest pleas to keep chains of thought monitorable for

AI safety (Korbak et al., 2025). This skepticism seems warranted given that there is no theory drawing causal connections between the semantics of the intermediate tokens and the final solution–beyond the usual "intermediate tokens change the conditional distribution of the solution tokens." After all, even the proponents of user interpretable semantics for intermediate tokens seem to realize this and write papers talking about why it is incumbent to protect the fragile connection between intermediate tokens and final solutions so as to allow for some kind of monitorability of LLMs (even if it may well be only illusory). Until there is evidence–beyond circumstantial–that LRM reasoning tokens correspond to internal computations, it seems unwise to depend on the intermediate tokens for "safety monitoring" in safety-critical domains. A third-party verification of the solutions/decisions (Kambhampati et al., 2024) seem to be the better way to go in such scenarios.

Our results from Section 7 imply that the prevalent practice of viewing the length of intermediate tokens as a measure of the computational complexity of the problem instance, might not necessarily be correct. We believe this conflation happened largely because the frontier models charge the end users based on the length of output tokens, a large fraction of which are intermediate tokens in reasoning models. We elaborate on all these concerns and recommendations in a recent position paper (Kambhampati et al., 2026).

In terms of explaining the role of intermediate tokens in the performance of current reasoning models, our main contribution is to urge the community to look beyond semantics of the reasoning traces, or their correlation to problem complexity. One speculative future direction that we believe merits investigation is viewing intermediate tokens as prompt augmentations (Kambhampati et al., 2025). The intuition is that for a given task prompt $T$, there may exist an augmentation PA which boosts the LLM's performance on that task:

$$\mathbb{P}\big(\text{Sol}\big(\text{LLM}(T + \text{PA}), T\big)\big) > \mathbb{P}\big(\text{Sol}(\text{LLM}(T), T)\big)$$

Here $\text{Sol}(y, T)$ indicates that $y$ solves $T$, and $\text{LLM}(x)$ is the model's completion for input $x$. The central challenge then is to learn the Skolem function

$$\text{PA} = f_\theta(T, \text{LLM}),$$

that maps each task to an effective augmentation. This can be accomplished through modifying the model itself to inherently and automatically augment prompts, as is the case in models that first generate long chains of intermediate tokens before their final answers. Crucially, prompt augmentations have no need to be human-interpretable. In fact, we see results that back this up in the adversarial prompting literature, where effective jailbreaks can be effected by augmenting prompts with human-uninterpretable strings (Zou et al., 2023; Cherepanova & Zou, 2024; Liu et al., 2024; Hackett et al., 2025) or modifying them with random syntactic permutations, capitalizations, and shufflings (Hughes et al., 2024), as well as the recent work on using intermediate tokens from the continuous latent space (Hao et al., 2024). In this prompt augmentation view, intermediate tokens (and their length) can perhaps be interpreted as the scaffold that the reasoning model uses to learn to manipulate it's context to bring the current instance closer to its training distribution, rather than necessarily a reflection of the computational hardness of the problem instance.

## Acknowledgments

This research is supported in part by grants from ONR (N00014-25-1-2301 and N00014-23-1-2409), DARPA (HR00112520016), DoD RAI (via CMU subcontract 25-00306-SUB-000), an IBM Research Fellowship to the first author and a generous gift from Qualcomm. We also thank Amazon AWS and Thinking Machines for compute awards. The paper benefited from feedback and discussions with several of our lab members, including Lucas Saldyt, Siddhant Bhambri, Durgesh Kalwar, Upasana Biswas and Soumya Samineni.

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

# A  Appendix

## A.1  Additional experiment details

For all our experiments, we trained the Qwen-2.5-0.5B decoder only models. We used a custom tokenizer with domain specific vocabulary which reduced the model parameters to around 380M. We optimized with AdamW ($\beta_1$=0.9, $\beta_2$=0.999) and applied a weight decay of 0.1528, a peak learning rate of 2.2758e-4, and 100 warm-up steps, all under bf16 precision. All randomness was controlled with fixed seeds. We train the models for 100k training steps when the dataset size is 50k and 250k training steps when the dataset size is 500k.

For GRPO, we use 16 as the sample size, 256 as the batch size and 0.01 as the entropy coefficient for both the models.

## A.2  Additional Seed Run for SF-style Model

To assess the robustness of our findings to random weight initialization, we conducted additional independent training runs of the SF-style models using a different initialization seed. All other experimental settings such as training data, optimizer configuration, batch-size, training steps, etc. were kept identical to the experiments reported in Table 3.

This experiment evaluates whether the trends observed in the main seed, particularly the equivalence in performance between the normal model and the swapped model, are stable under stochastic variation in initialization.

Table 6: SF-style model performance across seeds. Original denotes the models whose performance were reported at Table3. Seed 2 are the models trained on a different seed. Each cell shows *Plan Accuracy (%) / Trace Validity within valid plans (%)*.

| Model Type | Wilson | Kruskal | DFS | Drunkard | SF-style |
|---|---|---|---|---|---|
| Normal (SF-style, original) | 40.8 / 89.0 | 44.6 / 93.0 | 19.9 / 85.4 | 62.1 / 85.2 | 56.2 / 81.1 |
| Normal (SF-style, Seed 2) | 43.3 / 42.6 | 44.3 / 43.6 | 20.6 / 18.7 | 66.3 / 62.3 | 64.9 / 58.9 |
| Swapped (SF-style, original) | 45.8 / 0.0 | 51.0 / 0.0 | 29.0 / 0.0 | 95.4 / 0.0 | 89.1 / 0.0 |
| Swapped (SF-style, Seed 2) | 48.5 / 0.0 | 54.9 / 0.0 | 30.7 / 0.0 | 87.8 / 0.0 | 86.8 / 0.0 |

We observe that the swapped model continues to outperform the normal model of the same initialization seed across all distributions. We can conclude that the model performance of the normal and swapped models are statistically robust to different weight initializations.

## A.3  Robustness to Trace Shuffling Seed

In addition to varying the model initialization seed, we evaluated whether the behavior of the Swapped models was dependent on a specific random shuffle of the traces used to construct the training dataset. To this end, we trained the swapped model (SF-style) on a dataset containing identical maze problems but shuffled the A∗ search traces associated with them using a different random seed prior to training. All other experimental settings, such as weight initialization seed, optimizer configuration, learning rate schedule, batch size, and total training steps, etc., were kept identical to the experiments reported in Table 3.

Table 7: Swapped (SF-style) model trained with an alternative data shuffling seed. Each cell shows *Plan Accuracy (%) / Trace Validity within valid plans (%)*.

| Model Type | Wilson | Kruskal | DFS | Drunkard | SF-style |
|---|---|---|---|---|---|
| Normal (SF-style) | 40.8 / 89.0 | 44.6 / 93.0 | 19.9 / 85.4 | 62.1 / 85.2 | 56.2 / 81.1 |
| Swapped (SF-style) | 45.8 / 0.0 | 51.0 / 0.0 | 29.0 / 0.0 | 95.4 / 0.0 | 89.1 / 0.0 |
| Swapped (SF-style, Shuffle Seed 2) | 42.8 / 0.0 | 49.6 / 0.0 | 28.8 / 0.0 | 92.5 / 0.0 | 86.4 / 0.0 |

The swapped model trained on a dataset constructed using a different trace shuffle seed exhibits performance that is highly consistent with the original swapped model across all test domains. These results indicate that the observed behavior of the swapped model is robust to variations in the trace shuffling seed.

## A.4 Breakdown of responses as confusion matrices

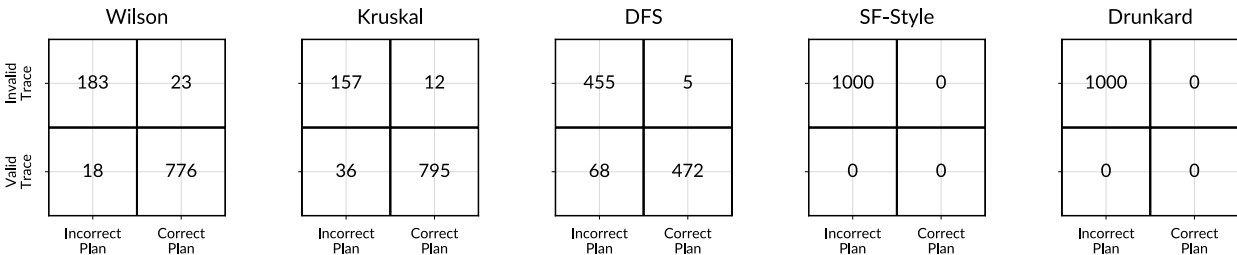

Figure 6: Plan validity versus trace validity for models trained on correct A* traces on 500k 30x30 Wilson mazes, measured across various maze problem distributions.



Figure 7: Plan validity versus trace validity for models trained on correct A* traces on 500k 30x30 Searchformer-style mazes, measured across various maze problem distributions.

We provide a detailed breakdown of model responses trained on mazes generated by Wilson's and SF-style generation. In section 4, we had provided the Plan accuracy and trace validity within responses that led to the valid plan. Here we present these results as confusion matrices in Figures 6 and 7, with each domain represented by a separate matrix. These results break down the correlation between model accuracy and trace validity. As can be seen from the results, trace accuracy is not a perfect predictor of plan accuracy. In fact, as can be seen from the diagonal entries, the model can produce valid traces and then continue on to produce an incorrect plan or produce invalid traces and yet end up at a correct plan.

### A.4.1 Error analysis of Model generated invalid traces

In this section, we provide a detailed break down of errors in model generated traces. Our evaluator, described in section 4, parses the model generated traces and flags them as incorrect if the trace violets any of the rules of A* search algorithm. We categorize execution errors into the following types: *Parsing Error*, *Wall exploration*, *Invalid Neighbor*, *Already Closed*, *Not in Open List*, *Trace-Plan mismatch* and *Goal Not Reached*. Detailed definitions of these error categories are provided in Section 4.

When a model generated trace is given to the evaluator, the function first checks if there are any syntax errors in the trace, i.e whether the trace is fully parseable. If the trace has a syntax error at any point, we categorize that response as Parsing error and not examine the trace further.

For semantically invalid traces, we report only the first encountered error. This is done because according to the rules of A* search, a single incorrect operation can cascade into multiple downstream errors (e.g., Adding a node that is not a valid neighbor of the node currently being expanded results in the evaluator flagging all subsequent operations performed on that node as invalid.). Reporting all the subsequent errors originated

from a single initial mistake would be misleading. To avoid the error-compounding effect, we report only the first encountered error.

Table 8 reports the first-error breakdown of responses generated by the models presented in Table 3. For the normal models, most of the invalid traces arise from parsing errors, followed by violations in which the model attempts to close a node that is not present in the open list. A non-trivial proportion of errors fall under the trace–plan mismatch category. In these cases, the derivational trace does not violate the procedural rules of $A^*$ search, yet the final plan produced by the model is inconsistent with its own reasoning trace. This indicates a lack of faithfulness between the intermediate derivational trace and the final answer.

In contrast, for the swapped models, 100% of the traces are invalid. Because the swapped dataset contains a trace corresponding to a different maze instance, the swapped model generated trace generally tries to expand a node other than the true start node. This results in the trace being flagged under the "close not in open list" category. Additionally, when the explored node corresponds to a wall cell in the current maze, the error is categorized as wall exploration. These patterns confirm that the swapped model systematically produces an reasoning trace corresponding to a different maze instance.

Table 8: First-error breakdown (%) of model-generated traces.

| Model | Parsing Error | Close $\notin$ Open | Trace Plan Mismatch | Wall Exploration | Invalid Neighbor | Goal Not Reached | Already Closed | No Error |
|---|---|---|---|---|---|---|---|---|
| Normal (Wilson) | 35.76 | 10.24 | 8.24 | 2.98 | 0.00 | 0.14 | 0.04 | 42.60 |
| Swapped (Wilson) | 10.10 | 59.38 | 0.00 | 30.52 | 0.00 | 0.00 | 0.00 | 0.00 |
| Normal (SF-style) | 31.12 | 18.00 | 6.72 | 3.34 | 0.08 | 0.78 | 0.24 | 39.72 |
| Swapped (SF-style) | 4.46 | 61.40 | 0.00 | 34.14 | 0.00 | 0.00 | 0.00 | 0.00 |

We also report the error-wise break down of responses reported in sections 4, 5 for normal models. For the Normal (Wilson) model, post-training slightly reduces procedural errors, leading to an increase in the proportion of valid traces ("No Error"). Parsing errors decrease modestly but remain the dominant failure mode throughout training.

In contrast, the Normal (SF-style) model, proportion of valid traces decreases with post training. While parsing errors decrease with post-training, errors associated with closing nodes not in the open list increase at later checkpoints.

Table 9: First-error breakdown (%) across post-training checkpoints for Normal (Wilson) and Normal (SF-style).

| Model | Checkpoint | Parsing Error | Close $\notin$ Open | Trace Plan Mismatch | Wall Exploration | Invalid Neighbor | Goal Not Reached | Already Closed | No Error |
|---|---|---|---|---|---|---|---|---|---|
| Normal (Wilson) | 0 | 35.76 | 10.24 | 8.24 | 2.98 | 0.00 | 0.14 | 0.04 | 42.60 |
| | 70 | 34.42 | 8.88 | 7.20 | 2.40 | 0.00 | 0.00 | 0.02 | 47.08 |
| | 140 | 34.36 | 7.92 | 7.84 | 2.04 | 0.00 | 0.00 | 0.00 | 47.84 |
| Normal (SF-style) | 0 | 31.12 | 18.00 | 6.72 | 3.34 | 0.08 | 0.78 | 0.24 | 39.72 |
| | 70 | 29.86 | 20.92 | 4.76 | 2.10 | 0.16 | 0.22 | 0.22 | 41.76 |
| | 140 | 21.74 | 40.18 | 3.80 | 1.24 | 0.04 | 0.12 | 0.04 | 32.84 |

## A.5 Validating Traces and Solutions of Searchformer Models

Along with our own trained models, we have also evaluated models trained by (Lehnert et al., 2024). These models have an encoder-decoder architecture and are trained on A* generated traces on 30x30 mazes [6]. The mazes are generated by their random generation method as described in Section 3. We see that across model sizes (from 15M to 175M parameters) there are a significant number of instances where the model produces a correct plan but the trace that it outputs is invalid. This is in line with the results of our models and provide further evidence that trace accuracy is not a perfect predictor of plan accuracy.

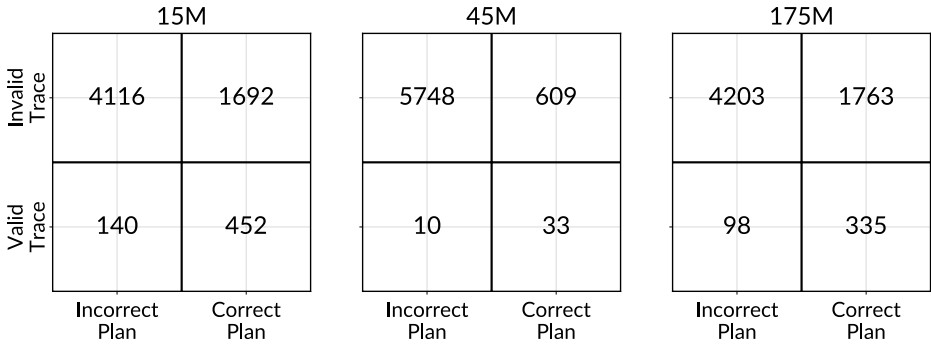

Figure 8: Plan validity versus trace validity for models trained on correct A* traces on 30x30 mazes, measured across varying model sizes and averaged over five runs (6400 responses per run).

## A.6 Evaluating Pre-trained LLMs

To evaluate whether our findings extend beyond models trained from scratch, we conducted additional experiments using a pre-trained large language model. Specifically, we fine-tuned Qwen3-8B-base on natural language derivational traces of the A* search algorithm across the three training regimes studied in the paper: *Solution-only*, *Normal (Correct traces)*, and *Swapped traces*.

### A.6.1 Model and Training Setup

- **Model –** We use Qwen3-8B-base as the base model.

- **Dataset –** The training dataset consists of 50k examples of $30 \times 30$ grid maze problems generated using the Wilson's algorithm. Each datapoint contains:
    - A maze specification (start, goal, and walls),
    - A natural language derivational trace of A* search (depending on the training baseline),
    - The final shortest path.

- **Finetuning Method and Hyperparameters-** We performed LoRA finetuning (Hu et al., 2022) with LoRA adapters applied to all linear layers of the model, with rank $r = 32$ and scaling factor $\alpha = 32$. Optimization was carried out using AdamW with ($\beta_1 = 0.9, \beta_2 = 0.95$). We employed a linear learning rate schedule with a peak learning rate of $4.72979 \times 10^{-4}$. Training was conducted for 25k steps with an effective batch size of 8.

### A.6.2 Direct Prompting and SFT evaluations

We follow the following baselines for our experiments -

1. **Direct Prompting -** We directly prompt the base model to solve the grid navigation problem and give the final plan as a tuple of coordinates.

---

[6]We found that the dataset used to train the models had <1% of instances with incorrect traces. Therefore we created our own A* implementation to ensure complete correctness of the traces within our generated datasets.

**Example prompt -**

```
You are given a 30x30 grid maze for A* search.
Start: (2,10)
Goal: (8,3)
Walls: [(5,0), (15,0), ...(29,29)]
Provide the final path as a single Python array of coordinate tuples.
```

2. **Vanilla SFT -** Following the conventional fine-tuning technique, we also utilize the SFT baseline where we fine-tune the models using only Input-Output pairs and no intermediate traces.

3. **SFT w/ correct traces -** To examine the impact and correctness of intermediate traces, we fine-tune the base model on A* traces followed by the correct plan. The training dataset has 50k problems generated using the Wilson's algorithm. Unlike the synthetic token-based traces used in earlier experiments, here we express the A* search traces in structured natural language. Since the trace is structured, we can construct a trace validator similar to the one which we had in section 4.

**Example of a Datapoint**

**Prompt -**

```
You are given a 30x30 grid maze for A* search.
Start: (21,18)
Goal: (10,0)
Walls: [(3,0), (7,0), (9,0).... (28,29), (29,29)]

Run A* and produce a derivational trace in natural language following these
↪   conventions:
When a node is added to the OPEN list, write:
"Adding Node with coordinates (x,y) in the open list with cost g=G and heuristic h=H."

When a node is added to the CLOSED list, write:
"Closing Node with coordinates (x,y) in the close list with cost g=G and heuristic
↪   h=H."

After the search, output the final path as a single Python array of coordinate tuples.
```

**Reasoning Trace -**

```
<think>
Closing Node with coordinates (16,16) in the close list with cost g=16 and heuristic
↪   h=0.
Adding Node with coordinates (16,15) in the open list with cost g=15 and heuristic h=1.
Adding Node with coordinates (16,17) in the open list with cost g=17 and heuristic h=1.
Adding Node with coordinates (15,16) in the open list with cost g=15 and heuristic h=1.
....
Adding Node with coordinates (8,8) in the open list with cost g=0 and heuristic h=24.
Closing Node with coordinates (8,8) in the close list with cost g=0 and heuristic h=24.
</think>
```

**Final Answer -**

```
<answer>
[(16,16),... (10,8), (9,8), (8,8)]
</answer>
```

4. **SFT w/swapped traces -** We construct the swapped training dataset similar to how we discussed in section 6. These datasets consists of the exact same problems and final paths as the original 50k dataset, but the traces will be shuffled within problems.

### A.6.3    Results

We evaluate each fine-tuned model on held-out test sets generated using the different maze generation algorithms. Since the traces are also structured, we can design a trace validator similar to the one which we had in section 4.

Table 10: Performance of Qwen3-8B-base under different fine-tuning regimes. Each cell shows *Plan Accuracy (%) / Trace Validity within valid plans (%)*.

| Model | Wilson | Kruskal | DFS | Drunkard | Searchformer |
|---|---|---|---|---|---|
| Base Model | 1.3 / 0.0 | 1.5 / 0.0 | 1.3 / 0.0 | 14.5 / 0.0 | 0.2 / 0.0 |
| Solution-only | 99.7 / 0.0 | 98.5 / 0.0 | 78.2 / 0.0 | 61.5 / 0.0 | 0.2 / 0.0 |
| Correct Traces | 100.0 / 100.0 | 99.9 / 100.0 | 99.3 / 100.0 | 53.4 / 33.7 | 6.0 / 6.6 |
| Swapped Traces | 98.0 / 0.0 | 98.2 / 0.0 | 76.6 / 0.0 | 38.1 / 0.0 | 0.7 / 0.0 |

Preliminary results in Table 10 suggest that our findings in the controlled setting can generalize to pre-trained models.[7] Notably, we observe that models trained with incorrect traces can achieve substantial gains in plan accuracy over the base model even when trace validity remains zero.

### A.7    GRPO on Models Trained with Varying Proportions of Swapped Traces

Table 11: Performance of 50k Wilson models with varying proportions of swapped traces. Each cell shows *Plan Accuracy (%) / Trace Validity within valid plans (%)*.

| Model / Checkpoint | Wilson | Kruskal | DFS | Searchformer |
|---|---|---|---|---|
| **Quarter-swap** | | | | |
| 0 | 64.7 / 84.9 | 67.9 / 88.2 | 44.5 / 73.5 | 0.3 / 0.0 |
| 70 | 80.3 / 93.0 | 82.9 / 93.4 | 47.5 / 81.3 | 0.3 / 0.0 |
| 140 | 82.5 / 91.9 | 86.7 / 92.5 | 51.6 / 79.9 | 0.2 / 0.0 |
| **Half-swap** | | | | |
| 0 | 61.8 / 81.2 | 56.7 / 77.0 | 42.0 / 71.7 | 0.3 / 0.0 |
| 70 | 80.9 / 80.7 | 76.5 / 77.0 | 56.2 / 67.4 | 0.3 / 0.0 |
| 140 | 87.9 / 73.6 | 87.5 / 70.4 | 64.9 / 57.0 | 0.2 / 0.0 |
| **Three-quarter-swap** | | | | |
| 0 | 60.0 / 30.3 | 58.0 / 31.2 | 41.6 / 28.4 | 0.2 / 0.0 |
| 70 | 88.3 / 15.7 | 88.7 / 17.8 | 63.5 / 13.2 | 0.2 / 0.0 |
| 140 | 91.2 / 13.2 | 91.6 / 13.4 | 64.0 / 9.8 | 0.2 / 0.0 |

We additionally observe that models trained on swapped or partially-swapped datasets can benefit from further refinement using post-training methods such as GRPO. We find that GRPO yields measurable gains in plan validity across distributions with these models, while trace validity reduces in most cases. In other words, GRPO improves the solutions themselves but does not induce the model to produce more meaningful or correct intermediate traces. This effect is consistent across the swap-spectrum checkpoints reported in Table 11, where plan accuracy continues to increase after GRPO despite trace validity remaining flat or decreasing. These results suggest that, even when models are trained on mixed- or noisy-trace data—which may occur in practice when traces are scraped from the wild or generated using STAR-like bootstrapping procedures Zelikman et al. (2022)—post-training optimization can still reliably boost performance, without requiring alignment between the traces and problem at hand.

---

[7]It is possible that the Qwen model has been already trained on maze-solving problems and it only needed to get the solution format right, thus having a very high accuracy in the solution-only paradigm too.

