$$\exists\, PA \text{ s.t. } \mathbb{P}\big(\mathrm{Sol}\big(\mathrm{LLM}(T + PA),\, T\big)\big) > \mathbb{P}\big(\mathrm{Sol}(\mathrm{LLM}(T),\, T)\big)$$

Here $\mathrm{Sol}(y, T)$ indicates that $y$ solves $T$, and $\mathrm{LLM}(x)$ is the model's completion for input $x$. The central challenge then is to learn the Skolem function

$$\mathrm{PA} = f_\theta(T, \mathrm{LLM}),$$

that maps each task to an effective augmentation. This can be accomplished through modifying the model itself to inherently and automatically augment prompts, as is the case in models that first generate long chains of intermediate tokens before their final answers. Crucially, prompt augmentations have no need to be human-interpretable. In fact, we see results that back this up in the adversarial prompting literature, where effective jailbreaks can be effected by augmenting prompts with human-uninterpretable strings (Zou et al., 2023; Cherepanova & Zou, 2024; Liu et al., 2024; Hackett et al., 2025) or modifying them with random syntactic permutations, capitalizations, and shufflings (Hughes et al., 2024), as well as the recent work on using intermediate tokens from the continuous latent space (Hao et al., 2024).

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

# A  Appendix

## A.1  Additional experiment details

For all our experiments, we trained the Qwen-2.5-0.5B decoder only models. We used a custom tokenizer with domain specific vocabulary which reduced the model parameters to around 380M. We optimized with AdamW ($\beta_1$=0.9, $\beta_2$=0.999) and applied a weight decay of 0.1528, a peak learning rate of 2.2758e-4, and 100 warm-up steps, all under bf16 precision. We train the models for 95k training steps. All randomness was controlled with fixed seeds. The training dataset size is 50k datapoints unless specified otherwise.

For GRPO, we use 16 as the sample size, 256 as the batch size and 0.01 as the entropy coefficient for both the models.

## A.2  Breakdown of responses as confusion matrices

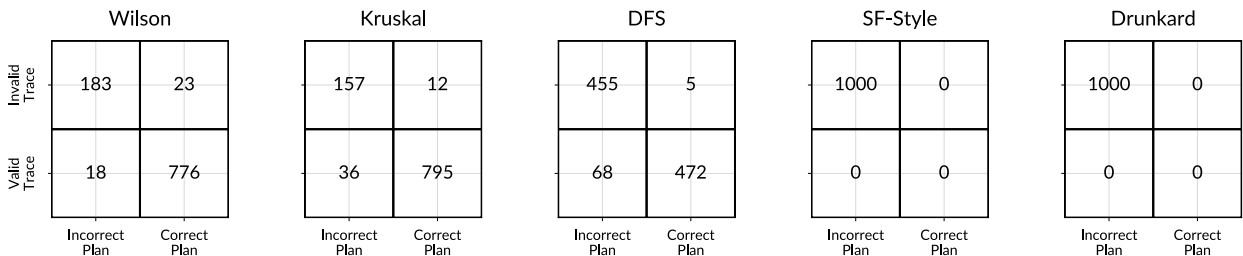

Figure 6: Plan validity versus trace validity for models trained on correct A* traces on 500k 30x30 Wilson mazes, measured across various maze problem distributions.

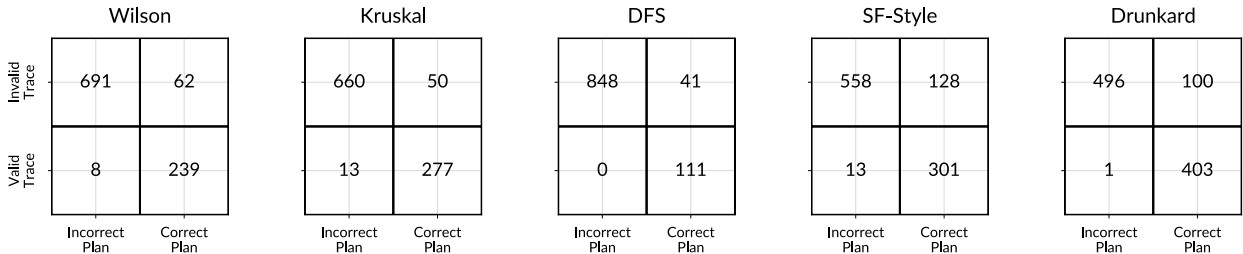

Figure 7: Plan validity versus trace validity for models trained on correct A* traces on 500k 30x30 Searchformer-style mazes, measured across various maze problem distributions.

We provide a detailed breakdown of model responses trained on mazes generated by Wilson's and SF-style generation. In section 4, we had provided the Plan accuracy and trace validity within responses that led to the valid plan. Here we present these results as confusion matrices in Figures [TO DO], with each domain represented by a separate matrix. These results break down the correlation between model accuracy and trace validity. As can be seen from the results, trace accuracy is not a perfect predictor of plan accuracy. In fact, as can be seen from the diagonal entries, the model can produce valid traces and then continue on to produce an incorrect plan or produce invalid traces and yet end up at a correct plan.

## A.3  Validating Traces and Solutions of Searchformer Models

Along with our own trained models, we have also evaluated models trained by (Lehnert et al., 2024). These models have an encoder-decoder architecture and are trained on A* generated traces on 30x30 mazes [5]. The mazes are generated by their random generation method as described in Section 3. We see that across model

---

[5]We found that the dataset used to train the models had <1% of instances with incorrect traces. Therefore we created our own A* implementation to ensure complete correctness of the traces within our generated datasets.

sizes (from 15M to 175M parameters) there are a significant number of instances where the model produces a correct plan but the trace that it outputs is invalid. This is in line with the results of our models and provide further evidence that trace accuracy is not a perfect predictor of plan accuracy.

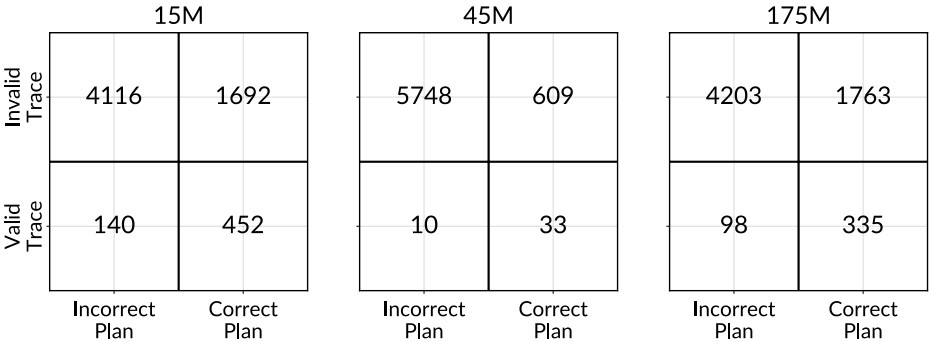

Figure 8: Plan validity versus trace validity for models trained on correct A* traces on 30x30 mazes, measured across varying model sizes and averaged over five runs (6400 responses per run).

## A.4 GRPO on Models Trained with Varying Proportions of Swapped Traces

Table 6: Performance of 50k Wilson models with varying proportions of swapped traces. Each cell shows *Plan Accuracy (%) / Trace Validity within valid plans (%)*.

| Model / Checkpoint | Wilson | Kruskal | DFS | Searchformer |
|---|---|---|---|---|
| **Quarter-swap** | | | | |
| 0 | 64.7 / 84.9 | 67.9 / 88.2 | 44.5 / 73.5 | 0.3 / 0.0 |
| 70 | 80.3 / 93.0 | 82.9 / 93.4 | 47.5 / 81.3 | 0.3 / 0.0 |
| 140 | 82.5 / 91.9 | 86.7 / 92.5 | 51.6 / 79.9 | 0.2 / 0.0 |
| **Half-swap** | | | | |
| 0 | 61.8 / 81.2 | 56.7 / 77.0 | 42.0 / 71.7 | 0.3 / 0.0 |
| 70 | 80.9 / 80.7 | 76.5 / 77.0 | 56.2 / 67.4 | 0.3 / 0.0 |
| 140 | 87.9 / 73.6 | 87.5 / 70.4 | 64.9 / 57.0 | 0.2 / 0.0 |
| **Three-quarter-swap** | | | | |
| 0 | 60.0 / 30.3 | 58.0 / 31.2 | 41.6 / 28.4 | 0.2 / 0.0 |
| 70 | 88.3 / 15.7 | 88.7 / 17.8 | 63.5 / 13.2 | 0.2 / 0.0 |
| 140 | 91.2 / 13.2 | 91.6 / 13.4 | 64.0 / 9.8 | 0.2 / 0.0 |

We additionally observe that models trained on swapped or partially-swapped datasets can benefit from further refinement using post-training methods such as GRPO. We find that GRPO yields measurable gains in plan validity across distributions with these models, while trace validity reduces in most cases. In other words, GRPO improves the solutions themselves but does not induce the model to produce more meaningful or correct intermediate traces. This effect is consistent across the swap-spectrum checkpoints reported in Table 6, where plan accuracy continues to increase after GRPO despite trace validity remaining flat or decreasing. These results suggest that, even when models are trained on mixed- or noisy-trace data—which may occur in practice when traces are scraped from the wild or generated using STAR-like bootstrapping procedures Zelikman et al. (2022)—post-training optimization can still reliably boost performance, without requiring alignment between the traces and problem at hand.