# OpenReview forum: "Beyond Semantics: The Unreasonable Effectiveness of Reasonless Intermediate Tokens"
_TMLR — Accepted by TMLR_

### Review · Reviewer_Jdjv · 2025-12-06

**Summary Of Contributions:**

The authors conduct a detailed study that evaluates the correlation between the validity of intermediate tokens (or chain-of-thought traces) with the overall performance of a transformer-based model. In particular, the authors first conduct a suite of controlled A*-based experiments that they use to systematically evaluate the validity of intermediate tokens, then use the findings from these experiments to try and answer several questions. Namely, the authors aim to determine whether the validity of intermediate tokens imply better performance, whether the presence of low-quality intermediate tokens inhibit performance, whether GRPO post-training improves the validity of intermediate tokens, and to what extent the (intermediate token) trace length correlates with the complexity of the problem being solved.

**Audience:**

Yes

**Audience Explanation:**

Given the prevalence of transformer-based models in the community, there would be a large audience for this paper.

**Claims And Evidence:**

No

**Claims Explanation:**

The authors make several claims in this paper:

1) That models trained on high-quality traces can still produce invalid (intermediate token) traces, while still arriving at correct solutions

2) That models trained on low-quality traces can achieve comparable or superior performance than models trained on high-quality traces.

3) That GRPO does not improve the validity of the traces even though the overall performance increases.

4) That the length of the trace is not correlated with the complexity of the problem that the model is trying to solve.

5) That the above claimed results challenge the assumptions of prior works that intermediate tokens reflect predictable human-like reasoning behaviors.

In terms of the first three claims, the authors do provide some evidence in support of their claims. In particular, the authors compare the Plan Accuracy (%) against the Trace Validity within Valid Plans (%) and make their conclusions based on these metrics. However, in my view, the evidence presented is not entirely convincing. In particular, the metric of *Trace Validity within Valid Plans (%)* is not motivated or justified, and, in my view, is too simple of a metric for what the authors are trying to show. Consider the following scenario: the model generates traces that are each 99% accurate (i.e., they make a mistake in, say, the final step and get the wrong solution). The reported metric for this case would be 0% / 0% (0% accuracy in the final solution and 0% of traces are fully valid). But is this 0% / 0% a fair characterization of the quality of the traces? I would argue that in this case the metric used by the authors fails to accurately capture the quality of the traces. As such, it would seem to me that, in general, having a binary *fully valid / not fully valid* metric (that is then aggregated into an overall percentage) seems too simple of a metric for what the authors are trying to show. Consequently, I do not find the first three claims sufficiently supported.

In terms of the fourth claim, the authors convincingly show that the length of the trace is not correlated with the complexity of the problem that the model is trying to solve, as shown in Figure 4.

Finally, in terms of the fifth and final claim, I do not see a clear and compelling argument presented. In particular, the authors consistently mention that prior works claim that traces are a reflection of coherent human-like thinking or reasoning, then cite several prior works that allegedly make this claim. Yet, not a single explicit example of such an argument is presented. As an analogy, think about the process of disproving a theory: first you state the theory/theorem that you want to disprove, then present proof/evidence that refutes the theory/theorem. However, in this paper, the authors claim that prior works make certain arguments but do not provide these exact arguments. Semantics matter here; what exactly did the prior works claim? What aspect of the claim is incorrect? Was the prior claim an actual claim or speculation? If the authors want to refute these claims, they need to be more precise and rigorous in how they go about it.

**Requested Changes:**

From a writing perspective, the paper is very well-written.

In terms of figures, Figure 2 has potential but is currently confusing to the reader. I think the figure would benefit from being presented as several subplots (a), b), c), etc.) that show the reasoning step by step and at which point the reasoning becomes incorrect. In terms of Figure 4, it is quite nonstandard to have a unity plot where the unity line is not presented at 45 degrees.

Non-formatting/grammar requests are as follows:
- The authors need to better motivate and justify the *Trace Validity within Valid Plans (%)* metric. Even better, I would strongly challenge the authors to consider whether this is truly the most compelling metric that they can use to support their claims.
- The authors need to be more precise about what prior claims they are refuting. Even better, it would greatly improve the narrative of the paper if the authors could present explicit examples of such claims, then use those claims to motivate the questions that they seek to answer in their paper.
- In Section 8, the authors, seemingly out of nowhere, introduce a lot of new speculation that should be removed from the paper. As an overall note, this paper feels a bit too loose in terms of speculation; I encourage the authors to evaluate whether all of this speculation is needed in their paper. In my view, it lowers the quality of the paper since many of these speculations are not rigorously supported.

---

> ### Author Response · Authors · 2026-02-19
>
> We thank the reviewer for their thoughtful comments. We address some of the concerns below:
>
> 1. **Motivation for the Trace Validity within Valid Plans (%) metric** -  Prior work that has looked at trace evaluations has only focused on partial and qualitative metrics in natural language settings [1,2,3], (in part because there is little else they could do with general natural language traces).  In contrast, here we wanted to provide a formally verifiable validity claim. Thus, we explicitly designed our experimental environment to include a formally verifiable trace, ensuring complete semantic validation of the traces rather than relying on qualitative or partial measures. This allowed us to provide precise guarantees regarding the validity of intermediate tokens. This stringent notion of trace correctness is directly aligned with the core question of whether trace correctness is tightly coupled to correct solutions and if it were, the statistic of Trace validity within valid plans would approach 100%-which it does not.
>
> However, in the revised manuscript, we have added an error wise breakdown of responses of Normal vs swapped vs post-trained models in the Appendix A.4.1. The responses are classified based on the first error encountered in the trace.
>
> We see that most of the traces generated by the swapped model are well formed and syntactically correct. However they get flagged as incorrect at the first step as the traces start from a different start state since they correspond to a different maze instance.
>
> For Normal models, we find that parsing error is the dominant cause of failure, followed by the procedural error of exploring a node that is not in the open list. We also observe that a non-trivial number of traces get flagged due to the Trace-Plan mismatch error which means that the model is not faithful to the reasoning it produced while producing the final plan.
>
> For Wilson based normal models, post-training increases the trace validity and thus we see a decrease in errors as training progresses. However, for SF-style based normal models, we observe an increase in violations of procedural rules of A* search with post-training and thus, the trace validity decreases as the training progresses.
>
>
> 2. **Explicit examples of prior claims being refuted** - Originally, in the introduction, we do say that our contention is with prior claims that have ascribed semantics to intermediate tokens and  specifically point out the ‘aha’ moment claim in the deepseek paper.
> Based on the reviewer’s recommendation, we have now explicitly added examples of prior claims in the introduction and the discussion and conclusion sections. These include claims in [2] that four key cognitive behaviors - verification, backtracking, subgoal setting, and backward chaining- are responsible for increasing solution accuracy with traces. In [1], the authors claimed that reasoning traces follow a consistent structure of problem formulation followed by repeated exploration and solution verification. These claims were based on qualitative metrics and no systematic verification of the said processes.
> The added text is marked in blue color.
>
> 3. **Speculation in section 8** - Our intention with the prompt augmentation hypothesis was to draw attention to a potential direction for future research in explaining the role of intermediate tokens of current reasoning models. Based on the feedback, we have now framed the hypothesis much more explicitly as a future direction to consider, demarcating it from our empirical contributions.
>
> 4. **Changes in figures** - Based on the reviewer’s comments, we have modified the caption in figure 2 to clarify the example trace and the possible errors. In Fig. 4, we intentionally do not plot the 45° unity line because it causes substantial clutter above the line near the left side of the plot and obscures the dispersion in response lengths. This is especially important given the large dynamic range: the maximum ground-truth A* search length on the x-axis is ~6k tokens, while model response lengths on the y-axis can reach ~32k tokens. Omitting the unity line improves readability and makes the scatter structure across the full range easier to interpret.
>
> References -
>
> [1] Sara Vera Marjanović, Arkil Patel, Vaibhav Adlakha, Milad Aghajohari, Parishad BehnamGhader, Mehar Bhatia, Aditi Khandelwal, Austin Kraft, Benno Krojer, Xing Han Lù, et al. Deepseek-r1 thoughtology: Let’s think about llm reasoning. arXiv preprint arXiv:2504.07128, 2025.
>
> [2] Kanishk Gandhi, Ayush Chakravarthy, Anikait Singh, Nathan Lile, and Noah D Goodman. Cognitive behaviors that enable self-improving reasoners, or, four habits of highly effective stars. arXiv preprint arXiv:2503.01307, 2025.
>
> [3] Samineni, Soumya Rani, et al. "Local Coherence or Global Validity? Investigating RLVR Traces in Math Domains." arXiv preprint arXiv:2510.18176 (2025).

---

> > ### Comment · Reviewer_Jdjv · 2026-02-20
> >
> > I thank the authors for their comments. I would say that my concerns have almost all been addressed by the changes made to the manuscript. However, I still do not see the motivation for the metric provided in the text. The authors should keep in mind that readers may have similar reservations about the metric as I do, and so the authors should incorporate their response (Motivation for the Trace Validity within Valid Plans (%) metric) in the main body.

---

> > > ### Author Response · Authors · 2026-03-03
> > >
> > > Thank you for your comments. We are grateful that you find the revisions and the justifications to your satisfaction. We will add the motivation provided for the Trace Validity within Valid Plans (%) metric in the main text of the final version of the paper.

---

### Review · Reviewer_nC4W · 2025-12-12

**Summary Of Contributions:**

This paper attempts to precisely answer the question of whether LLMs are using Chain-Of-Thought traces in order to “think through” a problem in a setting where this question can be precisely identified. Specifically, they investigate maze solving, where the solution is a “plan”, or a sequence of moves to get from the start to the goal state, and the thinking trace is the output of an A* algorithm run on the relevant graph. They then train models on trace/plan pairs and find that even in cases where models have perfect plans, the traces are often inaccurate, with trace accuracy among accurate plans falling when the model is evaluated off-distribution. They then train models with coherent traces that do not correspond to the eventual plan (i.e., traces corresponding to other similar instances of the same problem). They find that in general, doing this does not reduce (and in fact improves) performance. In a separate experiment, they find that having 50% traces correspond to the correct problem reduces performance relative to either 0% or 100%, suggesting that what is helpful is consistency rather than correctness. Results are similar for post-training GRPO rather than training. They also find that traces do not generally seem to have lengths corresponding to the true A* trace length, further reinforcing that traces are not “thinking” in the sense of following the relevant algorithm. They conclude that interpreting intermediate tokens produced by large language models as thinking is misleading and instead they should be thought of more simply as prompt augmentations, that are being jointly optimized with the prompt.

**Audience:**

Yes

**Audience Explanation:**

Yes, it provides compelling evidence in one direction regarding a major point of dispute within the community studying LLM COT.

**Broader Impact Concerns:**

None that I can see.

**Claims And Evidence:**

Yes

**Claims Explanation:**

The main claims of this paper are all supported by empirical experiments laid out in the body of the paper. However, the lack of error bars on these results, makes it harder to see if comparisons are valid. See Requested Changes for more.

**Requested Changes:**

Major:

In general, there is little to no discussion of how swapping traces might increase performance over having traces paired with plans. If anything, this is strictly reducing the amount of information in the dataset, as it is applying an informationally-destructive operation (randomizing paired into unpaired data) while not adding any additional information. The fact that this improves performance is not just surprising given standard hypotheses given chain-of-thought reflecting thinking tokens, but also given hypotheses regarding the role of training data in training of neural networks. A greater discussion of this finding, ideally with a hypothesis on what is going on, ideally with some empirical validation of this hypothesis, would strengthen this paper considerably.

Additionally, it is unclear which comparisons specifically can reliably be made between different numbers. Have experiments been run with multiple seeds so that confidence intervals can be established? If so, they should be reported. If not, consider performing these replicate experiments. [If this reveals that swapping is in fact not better than having traces aligned, this makes the previous requested change redundant, of course]


Minor:
In Section 5, I assume you meant to say that these models are trained on traces, it is not made clear enough that this is the case. You should clarify this specifically in the text.

For Figure 4 and 5, correlation values should be provided, preferably both pearson correlation and spearman correlation.

Nits, I’ve identified a few of these, but there probably exist others:

Missing a space at the bottom of page 8: “...um of training runs.The most…”
Similarly on page 11: “...trace lengths.If such a corr…”

---

> ### Author Response · Authors · 2026-02-19
>
> We thank the reviewer for their detailed feedback. We provide responses for the reviewer's comments below:
> 1. **Statistical Reliability** - In Appendix A.2, we have added analysis across a second seed run for the models trained on Searchformer dataset. Our results across these trials still demonstrate that the swapped model performs just as well as the model trained on correct traces, reinforcing that trace validity doesn’t seem to be  necessary for performance gains. In Appendix A.3, we have added analysis of an additional experiment where the model is trained on a dataset where the traces were  shuffled using a different swap seed. Those results also indicate that the performance of the swapped model is robust to the swapping seed. Due to cost and time constraints, we could not report results across more runs for wilson models. However, we would like to point out that our findings are consistent  across substantially different training conditions (a) two training paradigms (SFT and GRPO) and (b) models trained on two independently constructed corpora (Wilson and Searchformer). We have observed similar trends across these axes which reduces the likelihood that this effect is a brittle artifact of a specific hyperparameter.
>
>     We also assess whether our findings persist in pre-trained models by conducting an experiment using the Qwen3-8B base model. Experiment details and results have been reported in Appendix A.6. Consistent with our main results, swapping traces (i.e., training on structurally well-formed but problem-irrelevant traces) yields similar task performance to training on correct traces, suggesting that the phenomenon is not specific to from-scratch transformer training. In an independent related work, the authors in [1] have also demonstrated that similar conclusions hold for pre-trained models on natural language tasks such as Question-Answering.
>
>
> 2. **Discussion on swapping traces** - We would like to point out that we do present a hypothesis as to why swapping traces perform just as well as pairing traces with correct plans at the end of section 6 with empirical evidence which suggests that what matters for transformers to improve accuracy with intermediate tokens is not their semantic correctness, but rather their consistency. We also want to note that our motivation for swapping was only to show that there can exist perturbations in traces which destroy the correctness of the traces while still maintaining final task performance and bring into question the spurious correlations existing works draw between procedural correctness of thinking traces and solution correctness.
>
> 3. **Correlation values** - Based on the reviewers recommendation, we have now added the Pearson correlation and Spearman correlation values in section 7. The added text is marked in blue color.
>
> 4. **Section 5 correction and Nits** - We have clarified this in the text by adding “Each datapoint consists of a start and goal state, a maze definition, an A∗ trace representing the search process between the given start and goal states, and the corresponding correct plan.” We thank the reviewer for identifying the Nits, we have now fixed them.
>
> References -
>
> [1] Bhambri, Siddhant, Upasana Biswas, and Subbarao Kambhampati. "Interpretable traces, unexpected outcomes: Investigating the disconnect in trace-based knowledge distillation." arXiv preprint arXiv:2505.13792 (2025).

---

> > ### Comment · Reviewer_nC4W · 2026-02-20
> >
> > 1. Thank you for your update. I think the A2 and A3 results give me far more confidence. However, I think a sentence in A6 is needed describing how, while swapped traces underperform solution only in the finetuned setting, which is a different result from the  ones in the main paper.
> >
> > 2. That is acceptable, though it would be interesting to know why there's an *improvement* rather than a lack of reduction.
> >
> > 3. Thank you, this makes the results far clearer.
> >
> > 4. Thank you
> >
> > Please address the A6 issue above, otherwise all my concerns are allayed.

---

> ### Author Response · Authors · 2026-03-03
>
> Thank you for your comments. We are grateful that the revisions have addressed all your concerns. With regards to the A6 issue, we will add a sentence in the final version pointing out that in the preliminary results, swapped traces is underperforming compared to solution only in finetuned settings.

---

### Review · Reviewer_f4nK · 2026-02-04

**Summary Of Contributions:**

### Contributions

* Controlled study of CoT semantics: Authors train transformers from scratch in a *formal* domain (e.g. 30×30 maze shortest-path planning) where intermediate tokens are A* search traces and can be formally validated.
* Formal trace validator: defines trace validity by simulating the emitted A* operations and flagging errors (e.g., malformed actions, illegal neighbors, closing nodes not in open list, goal never closed).
* Most relevant empirical findings:
  1. **Weak coupling between correct answers and valid traces**: models can output a correct plan even when the trace is invalid.
  2. **Semantics of traces not necessary for performance gains**: models trained on *swapped (problem-irrelevant) traces* achieve accuracy comparable to (often better than) models trained on correct traces, while having **0% trace validity** (Table 3) and sometimes better OOD performance.
  3. **RL post-training (GRPO) improves plan accuracy but not trace validity**: accuracy rises while trace validity stays flat or even drops.
  4. **Trace length != problem difficulty / inference-time scaling**: generated trace length shows little correlation with ground-truth A* trace length; models may run to the context limit.
* **Interpretation/speculation**: proposes viewing intermediate tokens as a learned prompt augmentation (PA) improving $P[\text{Sol}(\text{LLM}(T+PA),T)]$ rather than as human-interpretable reasoning.

### strengths

* The formal A* domain and verifier let the paper cleanly separate "trace looks plausible" from "trace is correct."
* The swapped-trace construction is a strong test of end-user semantic alignment while keeping length/format similar to normal traces.
* Clear empircial story (baseline vs trace training vs swapping vs RL vs length analysis) with consistent measurement of trace validity.

### Weaknesses

* I think external validity is somewhat limited. By this I mean conclusions about "CoT semantics" for natural-language reasoning models are **suggestive**, but the evidence is from a single algorithmic domain and a single trace formalism.
* "Reasonless" feels a bit overstated, swapped traces are *task-mismatched* but still highly structured in-domain sequences; gains could reflect regularization / training dynamics / length/format effects, not purely "semantics don't matter."
* Some clarity/consistency issues. Training-step counts and dataset-size descriptions appear inconsistent across sections? (e.g. main-text long training vs Appendix A.1 shorter training regime), and Appendix A.2 contains a "[TO DO]" placeholder---these undermine polish.
* Novelty claim seems a bit too strong. In particular, "first to rigorously evaluate trace correctness" should be carefully scoped given broader neural algorithmic reasoning work that evaluates intermediate states/traces (even if not framed as CoT faithfulness).

**Audience:**

Yes

**Audience Explanation:**

The topic is relevant to the study of LLMs.

**Claims And Evidence:**

Yes

**Claims Explanation:**

The paper’s central claims are backed by:

* Controlled comparisons that change trace *semantic relevance* while holding trace *presence/format* (normal vs swapped) and showing comparable or better plan accuracy with swapped traces, while trace validity collapses to 0%.
* Formal validation, not heuristic grading, for trace correctness (Sec. 4).
* Multiple distributions / OOD tests (Wilson/Kruskal/DFS/Drunkard/SF-style) showing the pattern persists across environments.
* Post-training evidence that GRPO raises plan accuracy without improving trace validity.
* Length analysis directly comparing generated lengths to ground-truth A* trace lengths.

Main caveat: the paper sometimes **generalizes rhetorically** toward frontier LLM CoTs and "inference-time scaling" broadly; I think the presented evidence supports that direction *within this controlled setting*, but not as a universal statement.

**Requested Changes:**

### Most critical

1. **Tighten scope of claims (esp. title/abstract/discussion)**

   * Please consider rephrasing broad statements so they clearly read as: "In formally verifiable A*-trace maze planning…" rather than implying a general theorem about natural-language CoT in frontier models.
2. **Fix inconsistencies and placeholders**

   * Reconcile training-step/dataset-size descriptions across main text vs Appendix (e.g., 255k vs 95k-step regimes), and remove the "TO DO" in Appendix A.2.
3. **Report statistical reliability**

   * Add multiple-seed runs (or at least CIs/variance) for key comparisons where differences are small (e.g. normal vs swapped on in-distribution mazes), and for the headline OOD gains.
4. **Disentangle *why* swapped helps (minimal additional controls)**

   * At minimum, include one extra control that matches length but breaks *structure*, e.g.:

     * fixed dummy intermediate tokens,
     * token-shuffled traces,
     * random in-vocabulary trace-like strings.
       This is important because I think the paper's interpretation leans toward "consistency/format helps," but current evidence mainly shows "problem-alignment isn't required."

### Would improve the paper

1. **Broaden the controlled-study evidence**

   * Add a second formal domain (e.g., SAT traces, arithmetic procedure traces) or varied maze sizes to show the phenomenon isn't idiosyncratic to A* on 30×30 grids.
2. **Deeper error analysis**

   * Break down trace invalidity by error type (parse vs invalid neighbor vs open/closed list violations) for normal vs swapped vs post-trained models; this would clarify whether swapped models are "well-formed but wrong" vs "degenerate."
3. **Literature positioning**

   * Expand discussion around prior "..structure, not content.." findings (Li et al. 2025, they also cite some relevant work) and neural algorithmic reasoning/traces, and soften "first" claims accordingly.
4. **Optimality sensitivity check** (optional)

   * Since evaluation does not require optimal plans, add a small analysis of whether conclusions hold when scoring optimality (or at least report suboptimality rates).

---

> ### Author Response · Authors · 2026-02-19
>
> We thank the reviewer for their thoughtful comments. We address the reviewer's comments below:
>
> 1. **Statistical Reliability** - In Appendix A.2, we have added analysis across a second seed run for the models trained on Searchformer dataset. Our results across these trials still demonstrate that the swapped model performs just as well as the model trained on correct traces, reinforcing that trace validity doesn’t seem to be necessary for performance gains. In Appendix A.3, we have added analysis of an additional experiment where the model is trained on a dataset where the traces were shuffled using a different swap seed. Those results also indicate that the performance of the swapped model is robust to the swapping seed. Due to cost and time constraints, we could not report results across more runs for wilson models. However, we would also like to point out that our findings are consistent across substantially different training conditions (a) two training paradigms (SFT and GRPO) and (b) models trained on two independently constructed corpora (Wilson and Searchformer). We have observed similar trends across these axes which reduces the likelihood that this effect is a brittle artifact of a specific hyperparameter.
>
> 2. **Tightening the scope of the paper** - While we agree that our current findings do not constitute general theorems, we believe that the evidence provided here does suggest that the conclusions might extend to pre-trained models. To assess whether our findings persist in pre-trained models, we conducted an experiment using the Qwen3-8B base model. Experiment details have been reported in Appendix A.6. Consistent with our main results, swapping traces (i.e., training on structurally well-formed but problem-irrelevant traces) yields similar task performance to training on correct traces, suggesting that the phenomenon is not specific to from-scratch transformer training. In an independent related work, the authors in [1] have also demonstrated that similar conclusions hold for pre-trained models on natural language tasks such as Question-Answering.
>
> 3. **Fixing Inconsistencies and placeholders** - We thank the reviewer for pointing out the inconsistencies and we have now fixed them, highlighted by the text in blue color.
>
> 4. **Literature positioning** - Regarding the reviewer’s suggestion to relate our work to neural algorithmic works: Traditionally Neural Algorithmic works have focused on how to induce a faithful representation of an algorithm into a Neural Network. These prior works have mostly focused on inducing graph algorithms into Graph Neural Networks (GNNs).
> While there are works which have examined intermediate states and traces for checking if the GNNs have learnt a faithful execution of the algorithm, to the best of our knowledge, there are no similar works in the context of autoregressive transformers. While evaluating if the transformer models have learnt a faithful version of A* search, we go beyond that to see if problem relevant traces matter to induce performance gains, what happens to trace validity with GRPO and how the length of the trace is correlated with problem complexity.
> We have expanded the paragraph of training transformers on traces in the related work section with discussion on prior works in neural algorithmic reasoning, see the changes in “blue” color.
>
> 5. **Deeper error analysis** - Based on the reviewer’s recommendation, we have added an error wise breakdown of responses of Normal, swapped and post-trained models in the Appendix A.4.1. The responses are classified based on the first error encountered in the trace.
> We see that most of the traces generated by the swapped model are well formed and syntactically correct. However they get flagged as incorrect at the first step as the traces start from a different start state since they correspond to a different maze instance.
> For Normal models, we find that parsing error is the dominant cause of failure, followed by the procedural error of exploring a node that is not in the open list. We also observe that a non-trivial number of traces get flagged due to the Trace-Plan mismatch error which means that the model is not faithful to the reasoning it produced while producing the final plan.
> For Wilson based normal models, post-training increases the trace validity and thus we see a decrease in errors as training progresses. However, for SF-style based normal models, we observe an increase in violations of procedural rules of A* search with post-training and thus, the trace validity decreases as the training progresses.
>
> References -
>
> [1] Bhambri, Siddhant, Upasana Biswas, and Subbarao Kambhampati. "Interpretable traces, unexpected outcomes: Investigating the disconnect in trace-based knowledge distillation." arXiv preprint arXiv:2505.13792 (2025).

---

### Decision · Action_Editor_K16d · 2026-03-15

**Recommendation:** Accept with minor revision

**Additional Comments:**

All three reviewers recommend Accept after a thorough rebuttal process. The paper is sound and presents interesting, counter-intuitive findings within a well-controlled experimental setting. The review process was productive: reviewers raised substantive concerns about statistical reliability, metric justification, and claim scope; the authors responded with additional seed experiments (A.2, A.3), error breakdowns (A.4.1), pretrained model validation (A.6), correlation values, and revised text.

There are some minor revisions required before final acceptance.

1. It would be better to incorporate Trace Validity metric motivation into the main text (in Section 4 or 5) instead of the rebuttal.

2. The mixed evidence of Qwen3-8B should be discussed in the main text (wapped traces underperform solution-only in the finetuned setting), which is important for correctly scoping the paper's claims.

3. This work is recommended to tighten the claim scope in the discussion. The paper's strongest contribution is as a controlled counterexample demonstrating that problem-aligned trace semantics are not necessary for performance in a formal algorithmic domain, where the CoT-answer relationship is explicitly qualified as suggestive extrapolations.

**Audience:**

Yes

**Audience Explanation:**

This work focuses on the relationship between the CoT and final answer in a simple setting, which is among the most actively debated topics in the LLM community. This paper provides rigorous controlled evidence and challenges the prevailing assumption that intermediate tokens must be semantically aligned with the problem to be useful. The findings are directly relevant to researchers working on LLM reasoning, CoT faithfulness and post-training with RL.

**Claims And Evidence:**

Yes

**Claims Explanation:**

To verify the relationship between the reasoning trace and the final accuracy, the paper presents a well-designed controlled study in the $A^*$ maze pathfinding domain, where reasoning traces can be formally verified. Then, they first verify their core intuition: solution accuracy (Reasoning CoT in LLMs) and trace validity (answer in LLMs) in the multiple maze distributions (Table 1) and difficulty levels (Table 2). After that, they conduct swapped-trace experiments (Table 3) with multiple random seeds and show that models trained on problem-irrelevant traces achieve comparable or superior plan accuracy (e.g., 95.4% vs. 62.1% on Drunkard mazes for SF-style models) while maintaining 0% trace validity.  They also analyze the influence of the post-training algorithm to support their results.